# Kondo screening in a Majorana metal

S. Lee [1,9], Y. S. Choi [2,9], S.-H. Do [3,9], W. Lee [1,4], C. H. Lee [5], M. Lee [6], M. Vojta [7], C. N. Wang [8], H. Luetkens [8], Z. Guguchia [8] & K.-Y. Choi [2] ✉

Kondo impurities provide a nontrivial probe to unravel the character of the excitations of a quantum spin liquid. In the $S = 1/2$ Kitaev model on the honeycomb lattice, Kondo impurities embedded in the spin-liquid host can be screened by itinerant Majorana fermions via gauge-flux binding. Here, we report experimental signatures of metallic-like Kondo screening at intermediate temperatures in the Kitaev honeycomb material $\alpha$-RuCl$_3$ with dilute Cr$^{3+}$ ($S = 3/2$) impurities. The static magnetic susceptibility, the muon Knight shift, and the muon spin-relaxation rate all feature logarithmic divergences, a hallmark of a metallic Kondo effect. Concurrently, the linear coefficient of the magnetic specific heat is large in the same temperature regime, indicating the presence of a host Majorana metal. This observation opens new avenues for exploring uncharted Kondo physics in insulating quantum magnets.

When a magnetic impurity is introduced into a metal, conduction electrons interact with the local magnetic moment. At temperatures below the so-called Kondo temperature, the impurity spin becomes effectively screened by the surrounding conduction electrons, creating a many-body entanglement cloud[1]. This Kondo effect brings about a reduction in the magnetic moment of the impurity spins and a drastic increase in resistivity. Beyond normal metals, the purview of Kondo physics has expanded into various materials, including quantum dots, graphene, topological insulators, and Weyl semimetals[2–6]. It is also envisioned that the Kondo effect may occur in quantum spin liquids (QSLs) that constitute highly entangled quantum states harboring fractionalized spinon excitations, an emergent gauge structure, and topological order[7–16]. In addition, magnetic impurities incorporated into QSLs may be subject to RKKY-type interactions mediated by spinons or gauge fluctuations. In this context, Kondo impurities can act as in-situ probes for QSLs.

A $S = 1/2$ Kitaev model on the honeycomb lattice offers an archetypical platform for exploring unusual Kondo effects: While magnetic insulators often feature bosonic excitations, such as triplons or magnons, which cannot easily screen impurity spins, the fractionalized Kitaev QSL state hosts charge-neutral fermionic excitations, which can

effectively screen impurity spins. At finite temperatures $T$, itinerant Majorana fermions (MFs) wander around thermally activated $\pi$-fluxes ($W_p = -1$)[17,18], emulating metallic behavior, whereas the fluxes freeze out at low $T$, resulting in a Majorana semimetal. When a spin-1/2 impurity is exchange-coupled to a Kitaev spin, a first-order transition takes place at low $T$ as a function of the Kondo coupling between the weak-coupling flux-free phase and the strong-coupling impurity-flux phase. In the latter, each impurity moment binds a gauge flux in the enlarged impurity plaquette, thereby inducing locally metallic behavior of the MFs, in turn leading to Kondo screening[9–11].

A conspicuous candidate material for testing the proposed Kitaev Kondo effect is $\alpha$-RuCl$_3$[19,20], as it is in close proximate to a Kitaev QSL. Its spin Hamiltonian is best described by the $K$-$J$-$\Gamma$-$\Gamma'$ model

$$H = \sum_{\langle ij \rangle x} S_i^x \begin{pmatrix} J+K & \Gamma' & \Gamma' \\ \Gamma' & J & \Gamma \\ \Gamma' & \Gamma & J \end{pmatrix} S_j^x + \sum_{\langle ij \rangle y} S_i^y \begin{pmatrix} J & \Gamma' & \Gamma \\ \Gamma' & J+K & \Gamma' \\ \Gamma & \Gamma' & J \end{pmatrix} S_j^y$$
$$+ \sum_{\langle ij \rangle z} S_i^z \begin{pmatrix} J & \Gamma & \Gamma' \\ \Gamma & J & \Gamma' \\ \Gamma' & \Gamma' & J+K \end{pmatrix} S_j^z$$

[1]Center for Artificial Low Dimensional Electronic Systems, Institute for Basic Science, Pohang 37673, Republic of Korea. [2]Department of Physics, Sungkyunkwan University, Suwon 16419, Republic of Korea. [3]Materials Science and Technology Division, Oak Ridge National Laboratory, Oak Ridge, Tennessee 37831, USA. [4]Rare Isotope Science Project, Institute for Basic Science, Daejeon 34000, Republic of Korea. [5]Department of Physics, Chung-Ang University, 84 Heukseok-ro, Seoul 06974, Republic of Korea. [6]National High Magnetic Field Laboratory, Los Alamos National Laboratory, Los Alamos, New Mexico 87545, USA. [7]Institut für Theoretische Physik, Technische Universität Dresden, 01062 Dresden, Germany. [8]Laboratory for Muon Spin Spectroscopy, Paul Scherrer Institute, Villigen PSI 5232, Switzerland. [9]These authors contributed equally: S. Lee, Y. S. Choi, S.-H. Do. ✉e-mail: choisky99@skku.edu

with dominant Kitaev interaction $K = -5$–$10$ meV over Heisenberg ($J = -3$ meV) and off-diagonal symmetric exchange interactions $\Gamma = 2$–$3$ meV and $\Gamma' = 0.1$ meV[21–23]. $\alpha$-RuCl$_3$ shows the zigzag magnetic order below $T_N = 6.5$ K, preempting a Kitaev QSL. Although deviations from an ideal Kitaev model occur due to the presence of non-Kitaev terms, many independent experimental techniques suggest that Majorana and gauge degrees of freedom provide a good description of the $\alpha$-RuCl$_3$ magnetism at elevated energies and temperatures ($T > J$, $\Gamma$, $\Gamma'$)[22–25]. In addition, $\alpha$-RuCl$_3$ benefits from the availability of its isostructural counterpart CrCl$_3$ (Cr: $3d^3$; $S = 3/2$)[26,27]. CrCl$_3$ is a quasi-two-dimensional ferromagnet (FM) with consecutive FM and AFM orders at $T_C = 17$ K and $T_N = 14$ K, respectively. Taken together, mixed-metal trihalides $\alpha$-Ru$_{1-x}$Cr$_x$Cl$_3$ with random Ru/Cr occupancies[28] constitute a suitable model system for studying a Kitaev Kondo problem, gaining a fundamental understanding of $S = 3/2$ impurities embedded in a Kitaev paramagnetic host.

Here, we find several key signatures of metallic Kondo screening in a Kitaev paramagnetic state: logarithmic singularities in magnetic susceptibility, the muon Knight shift, and the muon spin-relaxation rate. Along with these characteristic Kondo signatures, a substantial magnetic contribution to the specific heat, $C_m/T$, raises the possibility that the observed Kondo screening arises from a Majorana metal host.

## Results

### Fractionalized spin excitations and structural homogeneity

Figure 1a schematically illustrates the formation of impurity plaquettes ($W_I = -1$; gray polygons) with binding of a gauge flux in the three adjacent plaquettes when $S = 1/2$ magnetic impurities are introduced to a Kitaev spin system. In Fig. 1b, we plot the $T$–$x$ phase diagram of $\alpha$-Ru$_{1-x}$Cr$_x$Cl$_3$ ($x = 0$–$0.07$), which reveals a slight reduction in the magnetic ordering temperature to $T_N \approx 5$ K. Additionally, within a Kitaev paramagnetic regime, there is an indication of a weak Kondo coupling, which is a central focus of this study.

We first confirmed the phase purity and composition of $\alpha$-Ru$_{1-x}$Cr$_x$Cl$_3$ through EDX and X-ray diffraction (XRD) analyses, as presented in Supplementary Figs. 1–3. Subsequently, we examine their

structural and magnetic excitations as a function of Cr$^{3+}$ impurity concentration $x$ to clarify the effects of the Cr-for-Ru substitution. Figure 1c shows the Raman spectra obtained at $T = 5$ K in in-plane polarization. For all the investigated $x = 0$–$0.07$, we observe a broad magnetic continuum (color shadings) with well-defined phonon peaks (Supplementary Fig. 4). In a Kitaev spin liquid, a magnetic Raman scattering process mainly involves the simultaneous creation or annihilation of pairs of MFs[29–31]. The observed magnetic Raman response comprises both MF and incoherent magnetic excitations, consistent with previous Raman data[27,29]. Remarkably, the magnetic continuum varies little with $x$ in its spectral form and intensity (the inset of Fig. 1c). The robustness of fractionalized excitations against Cr$^{3+}$ substitution indicates that a Kitaev paramagnetic state is hardly affected by the insertion of magnetic impurities. Moreover, the Cr$^{3+}$ substitution for Ru$^{3+}$ does not result in any essential changes in the frequency, FWHM, normalized intensity, and the asymmetry parameter $1/|q|$ of the $A_g(1) + B_g(1)$ and $A_g(2) + B_g(2)$ Fano resonance modes (Supplementary Figs. 4 and 5). Additionally, we could not detect any additional phonon peaks within the studied composition range. This observation, in conjunction with the absence of noticeable peak splitting in the single-crystal XRD data (Supplementary Figs. 2 and 3), strongly supports symmetry preservation, excluding the possibility of structural domains or phase segregation. These results suggest that the substituted Cr spins are randomly distributed throughout the lattice, although atomic-scale inhomogeneities cannot be entirely ruled out.

### Magnetic impurity effects on a static magnetic response

The Cr$^{3+}$-for-Ru$^{3+}$ substitution modifies the $K$-$J$-$\Gamma$-$\Gamma'$ exchange parameters of the mother compound $\alpha$-RuCl$_3$ by generating Heisenberg-type interactions on the Cr-Ru bonds. This is because Cr$^{3+}$ ions in the high-spin $d^3$ $S = 3/2$ configuration are orbitally inactive and, thus, are unable to provide multiple anisotropic and spin-dependent exchange paths required for $K$-$\Gamma$ interactions. In the Kitaev paramagnet, this changes the local energetics of the fluxes and also leads to the scattering of the itinerant MFs.

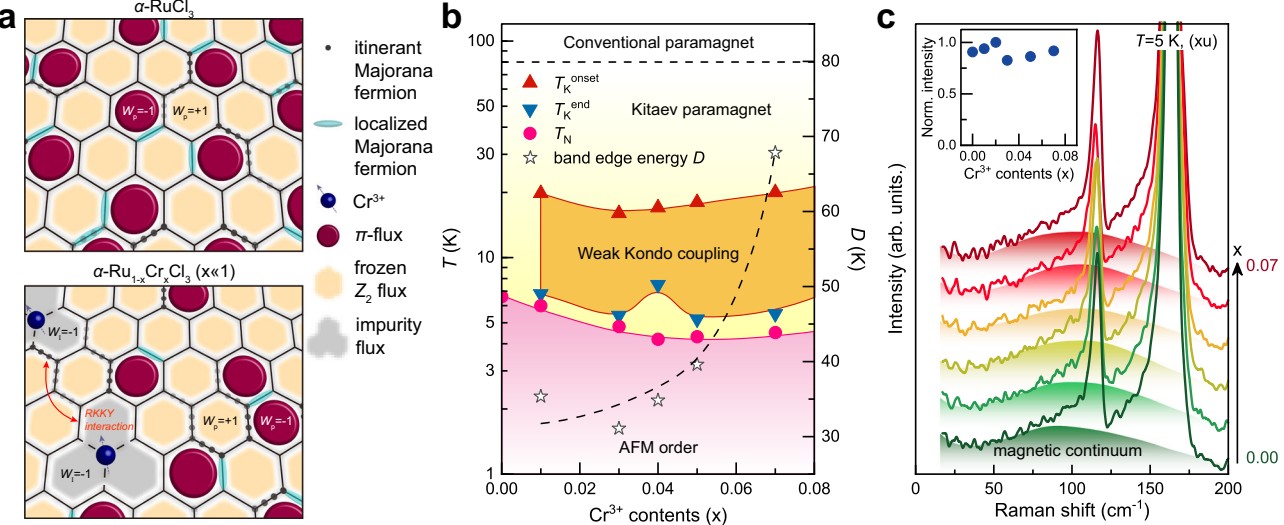

**Fig. 1 | Schematic sketch of gauge-flux-driven Kondo screening, $x$-$T$ phase diagram, and fractionalized excitations of $\alpha$-Ru$_{1-x}$Cr$_x$Cl$_3$. a** (Top) A Kitaev paramagnetic state consists of coherently propagating Majorana fermions (black dots) and thermally populated $\pi$-fluxes ($W_P = -1$) out of the frozen $Z_2$ gauge fluxes (incarnadine hexagons; $W_P = +1$). (Bottom) Spin-1/2 impurities coupled strongly to individual host spins (blue spheres) engender impurity plaquettes ($W_I = -1$; gray polygons) by a gauge flux in the three adjacent plaquettes. In addition, distant magnetic impurities can interact via long-range interactions (orange arrows). **b** $T$–$x$

phase diagram of $\alpha$-Ru$_{1-x}$Cr$_x$Cl$_3$ ($x = 0$–$0.07$). The characteristic temperatures $T_K^{onset}$, $T_K^{end}$, and $T_N$ are determined from the dc magnetic susceptibility, specific heat, and $\mu$SR measurements. The band edge energy $D$ is evaluated from the logarithmic fits to the magnetic susceptibility. The black dashed curve is a guide to the eye. AFM stands for antiferromagnetically ordered phase. **c** As-measured Raman spectra at $T = 5$ K. The color shadings denote the broad magnetic continuum. The inset plots the normalized intensity of the magnetic continuum as function of the concentration of the Cr$^{3+}$($S = 3/2$) impurities.

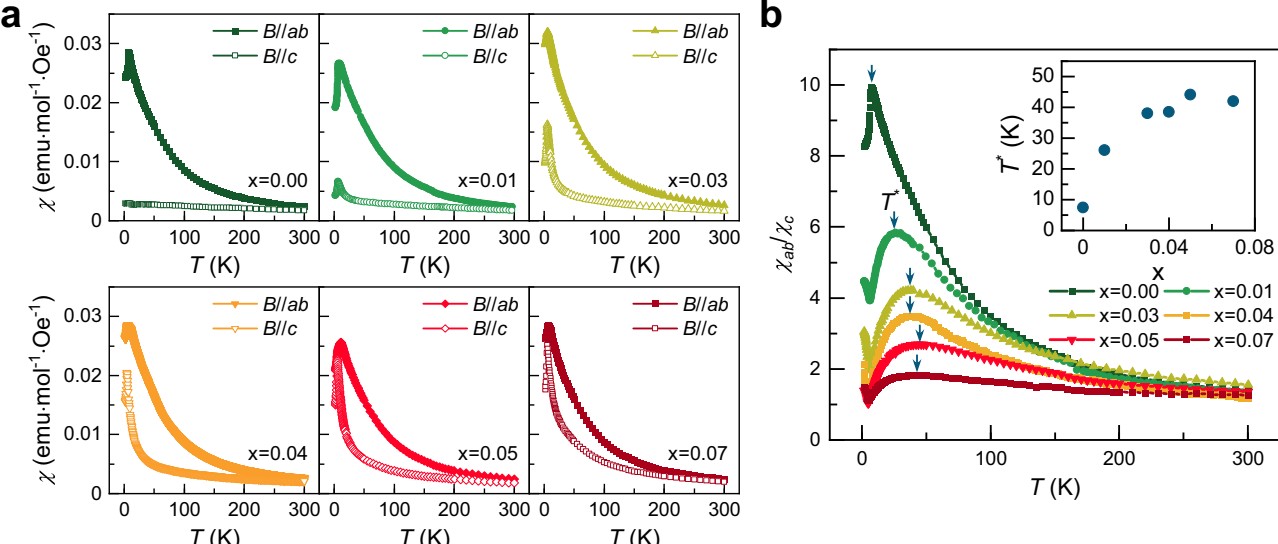

**Fig. 2 | Static magnetic susceptibility and magnetic anisotropy as a function of Cr content. a** Temperature dependence of dc magnetic susceptibility $\chi(T)$ of $\alpha$-$Ru_{1-x}Cr_xCl_3$ ($x = 0$–$0.07$) measured in an applied field of $B = 0.1$ T along the $ab$ plane (full symbols) and the $c$-axis (open symbols). The out-of-plane $\chi_c(T)$ shows a drastic increase with increasing $x$, rendering the magnetism of $\alpha$-$Ru_{1-x}Cr_xCl_3$ isotropic. **b** Temperature and composition dependence of the magnetic anisotropy $\chi_{ab}/\chi_c$ of $\alpha$-$Ru_{1-x}Cr_xCl_3$ measured in an applied field of $B = 0.1$ T. An XY-like magnetic anisotropy is systematically reduced with increasing $Cr^{3+}$ concentration. The downward arrows indicate the broad maximum temperature $T^*$ in $\chi_{ac}/\chi_c$. The inset plots $T^*$ versus $x$.

Figure 2a and Supplementary Figs. 6–8 exhibit the static magnetic susceptibilities $\chi(T)$ and magnetization of $\alpha$-$Ru_{1-x}Cr_xCl_3$ ($x = 0$–$0.07$) for $B//ab$ and $B//c$, along with corresponding Curie-Weiss fits. The Curie-Weiss behavior is identified in the paramagnetic state above $T = 100$–$180$ K (indicated by the dashed lines in Supplementary Fig. 6), and the Curie-Weiss parameters are summarized in Supplementary Fig. 7. The in-plane $\chi_{ab}(T)$ shows a small variation with $x$: the Curie-Weiss temperature $\Theta_{CW}^{ab}$ and the effective magnetic moment $\mu_{eff}^{ab}$ hardly change with increasing $Cr^{3+}$ impurities. The AFM ordering temperature is slightly reduced from $T_N = 6.5$ K at $x = 0$ to 5 K at $x = 0.03$–$0.07$ with no indications of spin-glass behavior down to 2 K. In sharp contrast to $\chi_{ab}(T)$, the out-of-plane $\chi_c(T)$ increases rapidly with increasing $x$. The large negative $\Theta_{CW}^c$ is drastically repressed towards $T = 0$ K and $\mu_{eff}^c = 3$ $\mu_B$ is reduced to 2.3 $\mu_B$ as $x$ increases up to 0.07 (Supplementary Fig. 7b, c). The drastic impact of the $Cr^{3+}$ impurities on $\chi(T,x)$ is quantified by the magnetic anisotropy $\chi_{ab}(T, x)/\chi_c(T, x)$, as shown in Fig. 2b. With increasing $x$, the XY-like magnetism becomes more isotropic, signaling that the $Cr^{3+}$ substitution weakens the $\Gamma$-$\Gamma'$ terms while enhancing the Heisenberg interaction[32]. Noteworthy is that a non-monotonic $T$ dependence of $\chi_{ab}/\chi_c$ features a maximum at about $T^* = 25$–$40$ K above $x = 0.01$ (the vertical arrows in Fig. 2b). The decrease of $\chi_{ab}/\chi_c$ below $T^*$ alludes to the growth of isotropic magnetic correlations beyond the underlying $K$-$J$-$\Gamma$-$\Gamma'$ magnetism.

### Logarithmic singularities of static magnetic susceptibility

A number of theoretical predictions have been made for impurities in Kitaev QSLs[9–11,33], but most of them are valid in the limit of low temperatures only. Here, we are interested in a *finite-T* crossover regime where conventional metallic-like Kondo screening would lead to a logarithmic increase of $\chi(T)$-$\ln(D/T)$, while the flux-binding mechanism in a semimetal would not lead to such logarithmic behavior[11].

To test the aforementioned scenarios, we plot $\chi_c(T)$ in Fig. 3a on a semilogarithmic scale, revealing a suggestive logarithmic behavior. To isolate the contribution induced by impurity spins, we present the difference of the static susceptibilities between the pristine and the $Cr^{3+}$-substituted samples, $\Delta\chi_c(T) = \chi_c(T)$-$\chi_c(T; x = 0)$ in Fig. 3b and Supplementary Fig. 9. Remarkably, we observe that $\Delta\chi_c(T)$ follows a logarithmic dependence, $\ln(D/T)$, in the temperature interval between $T_N$

and ~20 K. Within this range, we identify two characteristic temperatures, $T_K^{onset}$ and $T_K^{end}$, which delineate the interval where the logarithmic temperature dependence of $\Delta\chi_c(T)$ appears. In the $T = 30$–$100$ K range, the logarithmic $T$ dependence transits to an approximate power-law dependence $\chi(T) \sim T^{\alpha(T)-1}$ with $\alpha(T) \approx -0.12$–$0.14$ (Supplementary Figs. 9–11), which we interpret as a crossover to the high-$T$ Curie-Weiss-like regime. The deviation from $\alpha = 0$ is attributed to scatterings off of itinerant MFs by $Cr^{3+}$ impurities. The fit parameter $D$ is evaluated to be $D = 23$–$67$ K (the star symbols in Fig. 1b), which is comparable to the strength of the subdominant $J$-$\Gamma$-$\Gamma'$ interactions and roughly agrees with $T^*$ in Fig. 2b. These results suggest that $\alpha$-$Ru_{1-x}Cr_xCl_3$ displays Kondo physics different from the flux-driven mechanism of ref. 11. The out-of-plane $\chi_{ab}(T)$ data also hold logarithmic signatures, yet their weak $x$ dependence (Supplementary Fig. 12a) disallows extracting reliable parameters. Further, we note that the Kondo temperature cannot be tracked as the logarithmic behavior is disrupted by the onset of AFM order. Furthermore, we attempted to analyze the $\Delta\chi_c(T)$ data in terms of the equivalent three-channel Kondo model[34]. We observe a qualitative agreement within the temperature range of $T_N < T < T_K^{onset}$, but not extending to temperatures $T_K^{onset} < T$ (Supplementary Fig. 10). Moreover, the derived Kondo temperature $T_K$ is notably lower than $T_K^{onset}$. This discrepancy is related to the fact that $\Delta\chi_c(T)$ continues to increase upon cooling in the fitting range above $T_N$ (see Supplementary Fig. 10f) and that the Cr impurity in $\alpha$-$Ru_{1-x}Cr_xCl_3$ is described by a $S = 3/2$ inequivalent three-channel Kondo model[11], as detailed in Supplementary Note 3. In addition, the remaining deviations may originate from inadequate fitting functions and the influence of vison dynamics.

### Metallic behavior of Majorana fermions

To probe the $Cr^{3+}$ substitution effect on low-energy excitations, we examine the magnetic specific heat $C_m(T)$ obtained by subtracting lattice contributions from the total specific heat $C_p(T)$ (Supplementary Figs. 11 and 12 and "Methods"). In Fig. 3c, we compare $C_m(T)$ between $\alpha$-$Ru_{1-x}Cr_xCl_3$ ($x = 0.04$) and the pristine sample ($x = 0$). $C_m(T)$ of the $x = 0$ sample shows a $\lambda$-like peak at $T_N = 6.5$ K, followed by a plateau in the temperature range of $T = 15$–$50$ K and a subsequent increase up to $T_H \sim 100$ K. Upon introducing the $Cr^{3+}$ impurities, two weak anomalies

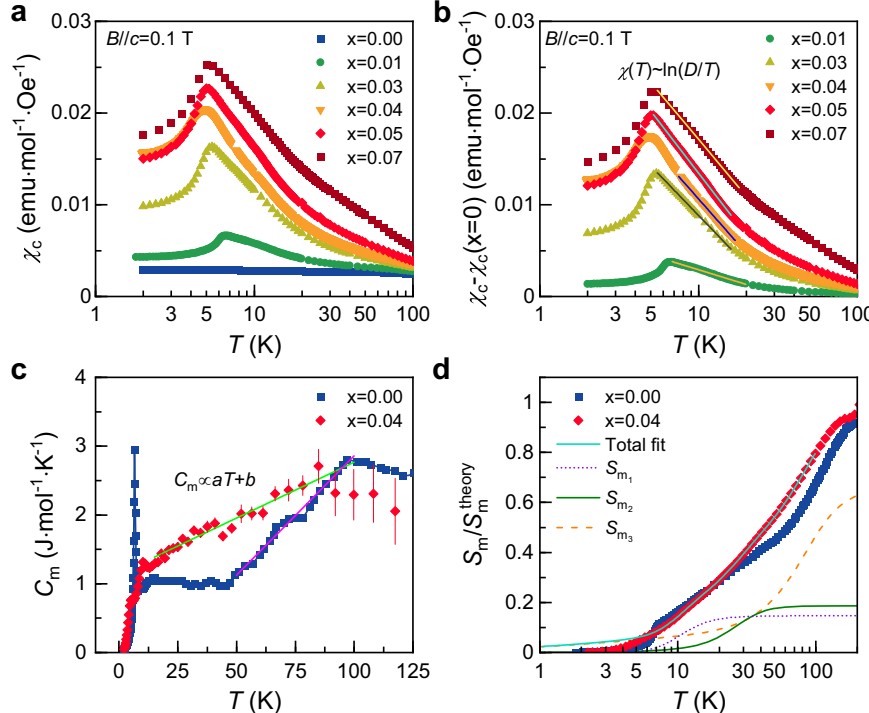

**Fig. 3 | Thermodynamic signatures of Kondo screening. a, b** Temperature dependence of the static magnetic susceptibility $\chi_c(T)$ and the pristine-subtracted $\Delta\chi_c(T) = \chi_c(T) - \chi_c(T;x=0)$ for $\alpha$-Ru$_{1-x}$Cr$_x$Cl$_3$ ($x = 0.01$–$0.07$) in an applied field of $B//c = 0.1$ T. The solid lines are fittings to logarithmic divergence $\Delta\chi(T) \sim \ln(D/T)$, where $D$ is the band edge energy. **c** Comparison of the $T$-dependent magnetic specific heat $C_m(T)$ between $\alpha$-Ru$_{1-x}$Cr$_x$Cl$_3$ ($x = 0.04$) and the pristine material ($x = 0$). $C_m(T)$ is obtained by subtracting a lattice contribution from the total specific heat (Supplementary Fig. 12). The solid lines indicate a $T$-linear dependence of $C_m(T)$. The error bars represent one standard deviation of the three repeated specific-heat measurements. **d** Normalized magnetic entropy $S_m/S_m^{theory}$ as a function of temperature evaluated by integrating $C_m(T)/T$ in a semi-log scale. $S_m^{theory}$ is $R$ln2 and $0.96R$ln2 + $0.04R$ln4 for $x = 0.00$ and $0.04$, respectively. The solid and dashed lines denote a fit using three phenomenological functions ("Methods").

appear at $T_{N1} = 4.8$ K and $T_{N2} = 10.4$ K for $x = 0.04$, corresponding to the magnetic ordering of ABC- and AB-type stacking patterns (Supplementary Figs. 11 and 12). As evident from Supplementary Fig. 11b, the addition of 2% magnetic impurities induces a linearly increasing fraction of $C_m$ in the intermediate $T = 13$–$50$ K plateau regime for $x = 0$. This trend is enhanced with increasing $x$ up to 0.04. The emergence of a linear $T$ contribution to $C_m$ below $T_H$ is a signature of metallic behavior of the itinerant MFs[35]: Such effective metallicity arises from the presence of thermally populated $\pi$-fluxes ($W_p = -1$), as illustrated in Fig. 1a.

Shown in Fig. 3d is the magnetic entropy $S_m(T) = \int C_m/T dT$. We recall that in an ideal Kitaev system, each half of $S_m(T)$ is released by itinerant and localized MFs[36]. Unlike the $x = 0$ sample[23], the magnetic entropy of $x = 0.04$ is released in three steps with the weighting factors $\rho_1 = 0.15R$ln2, $\rho_2 = 0.19R$ln2, $\rho_3 = 0.66R$ln2 ($R$ = ideal gas constant) and the crossover temperatures $T_1 = 10.7(3)$ K, $T_2 = 24(4)$ K, and $T_3 = 70(7)$ K ("Methods"). $T_1$ and $T_2$ correspond to the end temperatures where the logarithmic behavior of $\chi_c(T)$ appears (Fig. 1b). On the other hand, the power-law dependence $\chi(T) \sim T^{\alpha(T)-1}$ is observed between $T_2$ and $T_3$ (Supplementary Fig. 9). We note that one Kondo $S = 3/2$ spin is coupled to the three adjacent $S = 1/2$ sites, leading to flux conservation in the Kitaev QSL only in the joint six-plaquette area surrounding to the impurity[9–11]. Therefore, 4% Cr$^{3+}$ substitution modifies 24% of the fluxes near the impurities. Qualitatively, the three-step entropy release is consistent with this picture.

## Logarithmic singularities of the muon Knight shift and relaxation rate

To shine more light on the Kondo behavior, we carried out muon spin rotation/relaxation ($\mu$SR) measurements of $\alpha$-Ru$_{1-x}$Cr$_x$Cl$_3$ ($x = 0.04$) in zero (ZF), longitudinal (LF), weak (wTF), and high (hTF) transverse

fields. The wTF- and ZF-$\mu$SR data confirm the two successive magnetic transitions at $T_{N1} = 5$ K and $T_{N2} = 12$ K (Supplementary Figs. 13 and 14), in line with our magnetic and thermodynamic results.

As exhibited in Fig. 4a, the normalized fast Fourier transformed (FFT) amplitudes of the hTF-$\mu$SR spectra measured at $T = 15$ K show a Lorentzian shape with intriguing field evolution. Fittings reveal two Lorentzian relaxing cosine components (see Fig. 4b, c): (1) a sharp signal (yellow curve) and (2) a broad signal (green curve). The obtained fitting parameters are plotted in Fig. 4d–g and Supplementary Fig. 15. Given the fact that the field-induced crossover, involving the change of a magnetic domain structure, occurs across $B \sim 1$ T[37] (Supplementary Fig. 8), we chose the two representative fields $B_0 = 0.5$ and 3 T for detailed $T$-dependent studies.

The locally probed intrinsic magnetic susceptibility is reflected in the $T$-dependent muon Knight shifts $K_f(T)$ and $K_s(T)$. $K_s(T)$ and $K_f(T)$ scale well with $\pm\chi_c(T)$ down to 2 K (Supplementary Fig. 15), indicating that the logarithmic dependence of $\chi(T)$ seen in the $T = T_N -20$ K range is little affected by extrinsic contributions. Notably, $K_f(T)$ and $K_s(T)$ clearly show distinct temperature dependences. $K_s(T)$ displays a logarithmic dependence $\ln(D/T)$ above 10 K, while $K_f(T)$ shows a power-law behavior $T^{-n}$ (see Fig. 4d, e). The extracted $D = 30.8(6)$ K (27.9(7) K) for $B = 0.5$ T (3 T) is comparable to the value evaluated from the static $\chi_c(T)$ data shown in Fig. 3a, b. Furthermore, based on the relation $\lambda \sim 1/T_1 \sim A^2 T[\text{Im}\chi(T,\omega)/\omega]_{\omega \to 0}$, the muon relaxation rate could be expected to follow logarithmic behaviors of $\lambda \sim T[\ln(D/T)]^2$ for a single vacancy or $\lambda \sim 1/T[\ln(D/T)]^2$ for a pair of nearby vacancies on the same sublattice, respectively[9–11]. We find that only the slow relaxation rate $\lambda_s(T)$ for $B = 3$ T shows a logarithmic $T$ dependence $T[\ln(D/T)]^2$ with $D = 45(1)$ K. On the other hand, $\lambda_f(T)$ follows a power-law behavior $T^{-\alpha}$ with $\alpha = -0.98(1)$ for $B = 3$ T and $\alpha = -2.22(7)$ for $B = 0.5$ T above $T = 8$ K (see Fig. 4f). The concomitant power-law dependence of $\lambda_f(T)$ and $K_f(T)$

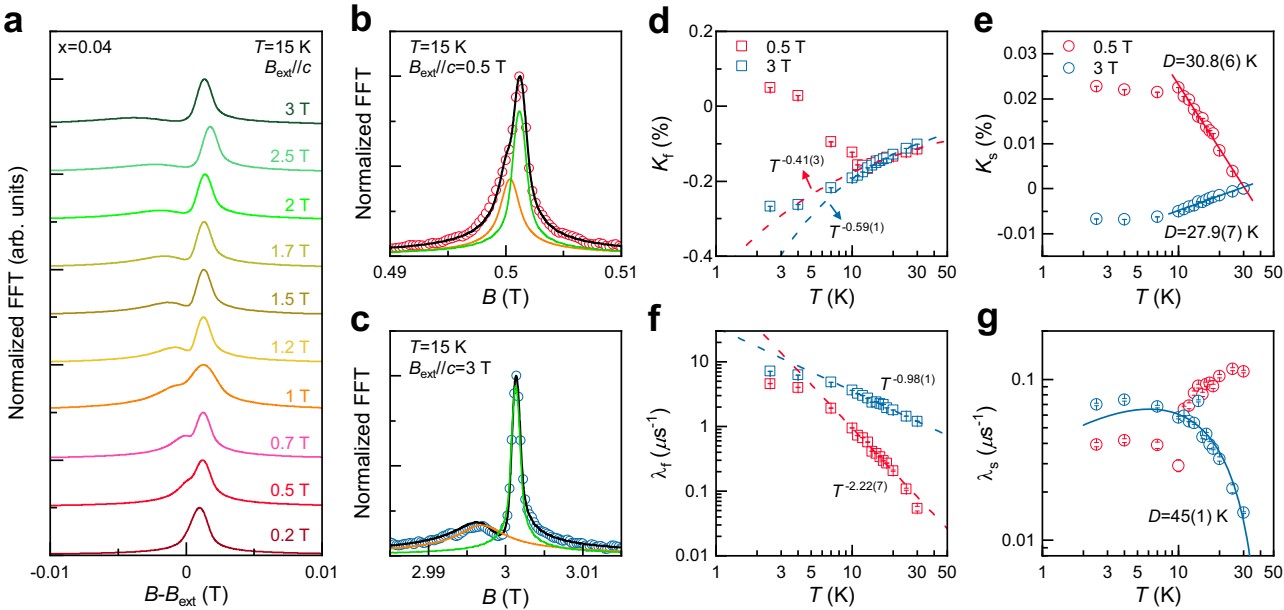

**Fig. 4 | High transverse-field $\mu$SR data of $\alpha$-Ru$_{1-x}$Cr$_x$Cl$_3$ ($x = 0.04$). a** Normalized FFT amplitudes of hTF-$\mu$SR in applied fields of $B_{ext}//c = 0.2$–3 T at $T = 15$ K. The data are vertically shifted for clarity. **b, c** Magnified views of normalized FFT amplitudes at $B_{ext} = 0.5$ and 3 T. The black solid lines denote the total fitting lines that are a sum of two Lorentzian damped cosines (yellow and green lines). **d, e** Temperature dependence of the muon Knight shift for the fast ($K_f$) and slow ($K_s$) relaxing components in applied fields of $B_{ext}//c = 0.5$ and 3 T. $K_f(T)$ is described by power-law behaviors $K_f \sim T^{-n}$ (dashed lines), which deviates below $T_{N2} = 12$ K, while $K_s(T)$ exhibits a logarithmic dependence $K_s$–ln($D/T$) (solid lines) predicted for a singlet vortex case above 10 K. Error bars represent one standard deviation. **f, g** Muon spin-relaxation rates for the fast ($\lambda_f$) and slow ($\lambda_s$) component as a function of temperature on a double logarithmic scale. $\lambda_f(T)$ displays a power-law down to $T_{N2}$ (dashed lines), similar to $K_f$. On the other hand, $\lambda_s(T)$ at $B_{ext} = 3$ T is well described by a logarithmic dependence $\lambda_s \sim 1/T_1 \sim T[\ln(D/T)]^2$ (solid lines). Error bars of the muon Knight shift and the relaxation rat represent one standard deviation of the fit parameters.

suggests that the fast component stems from correlated spins pertinent to defects and bond disorders, which inevitably occur due to stacking faults and local strains induced by the Cr$^{3+}$-for-Ru$^{3+}$ substitution. Actually, the static magnetic susceptibility follows an approximate power law $\chi(T) \sim T^{\alpha(T)-1}$ in the elevated temperatures of $T = 30$–100 K.

## Discussion

Combining specific heat, magnetic susceptibility, and $\mu$SR probes, we find that mixed-metal trihalides $\alpha$-Ru$_{1-x}$Cr$_x$Cl$_3$ offer a promising arena for exploring a Kitaev Kondo problem. The magnetism of $\alpha$-Ru$_{1-x}$Cr$_x$Cl$_3$ is modeled by the $K$-$J$-$\Gamma$-$\Gamma'$ spin Hamiltonian[32], where the strength of $J$ relative to $\Gamma$-$\Gamma'$ increases with $x$. Our findings reveal several key points.

First, we observe that the Cr$^{3+}$ substitution exerts no significant impact on fractionalized excitations at intermediate $T$ (Fig. 1c and Supplementary Fig. 4) despite the Heisenberg-type interaction $J_{Ru-Cr}$ perturbs the original $K$-$J$-$\Gamma$-$\Gamma'$ exchange interactions. Second, as evident from the rapid suppression of XY-like magnetic anisotropy in Fig. 2b, the inclusion of the spin-$\frac{3}{2}$ impurities diminishes the $\Gamma$-$\Gamma'$ terms, while augmenting the isotropic Heisenberg interaction. Third, $C_m(T)$ and $S_m(T)$, tracking thermal fractionalization of spins into itinerant MFs and $Z_2$ fluxes, demonstrate that the addition of magnetic impurities expands the Kitaev paramagnetic state down to $T_N$, which is much lower than ~50 K of $\alpha$-RuCl$_3$. The sizeable linear term in $C_m$, a hallmark of the metallic density of states, negates a paramagnon scenario. This expanded Majorana-metal regime can be rationalized by noting that the impurities both increase the fluctuations of the gauge fluxes and, at the same time, scatter the itinerant MFs, thereby inducing low-energy Majorana states. Fourth, both static and dynamic magnetic probes commonly feature logarithmic singularities of the conventional Kondo effect. Finally, the three-step release of $S_m(T)$, the three-step evolution of $\chi(T)$, and the magnetic anisotropy ($\chi_{ab}/\chi_c$) anomaly at $T \approx 25$–40 K

equivocally evidence the emergence of magnetic correlations induced by a few percentages of magnetic impurities.

This together with the large Kondo energy of ~30 K suggests that the scenario[11] of low-$T$ gauge-flux-driven Kondo screening in a Majorana semimetal is not applicable to $\alpha$-Ru$_{1-x}$Cr$_x$Cl$_3$. Instead, at elevated temperatures, a strongly fluctuating flux (or vison) background produces a Majorana metal host. In this situation, no explicit binding of fluxes to impurities is required for Kondo screening. Rather, the global presence of thermally excited gauge fluxes provides a natural mechanism for a metallic Kondo effect with logarithmic signatures, here for $S = 3/2$ moments with three inequivalent screening channels[11], here for $S = 3/2$ moments with three screening channels. At larger $x$, this Kondo physics will compete against the fluctuation-mediated inter-impurity interactions. We recall that the Kondo effect in a magnetic insulator has recently been reported in the Zn-brochantite ZnCu$_3$(OH)$_6$SO$_4$, a Kagome antiferromagnet that holds a proximate QSL[8]. In this case, magnetic impurities originating from Cu-Zn intersite disorders act as Kondo spins that may be screened by spinon-spinon interactions, but the precise mechanism has not been clarified. Thanks to its analytical solvability, however, an impurity-doped Kitaev system enables the exploration of uncharted territory including multi-channel Kondo physics and its interplay with gauge fluctuations.

To conclude, we have showcased metallic-like Kondo behavior in the Kitaev candidate material $\alpha$-Ru$_{1-x}$Cr$_x$Cl$_3$ containing $S = 3/2$ magnetic impurities, demonstrating the presence of a host Majorana metal. Multiple Kondo impurities and their interplay may bring about a new species of Kondo and ordering phenomena. Extending the present phenomena to low temperatures in a material without magnetic ordering would give access to the regime of flux binding by impurities[11], then raising the prospect of braiding impurity fluxes via impurity manipulation toward the implementation of quantum computation[17,18].

## Methods

### Sample preparation

Single crystals of $\alpha$-Ru$_{1-x}$Cr$_x$Cl$_3$ ($x = 0$–0.07) were synthesized by a vacuum sublimation method. A commercial compound of RuCl$_3$ (Alfa Aesar) was ground and dried in a quartz tube under vacuum until it was completely dehydrated. The resulting powder was then sealed in an evacuated quartz ampule, which was placed in a temperature gradient furnace. The ampule was heated at 1080 °C for 24 h and then slowly cooled down to 600 °C at a rate of 2 °C/h. The obtained single crystals have typical sizes of about $5 \times 5 \times 1$ mm$^3$ with a shiny black surface.

### Structural and thermodynamic measurements

The crystal structure of $\alpha$-Ru$_{1-x}$Cr$_x$Cl$_3$ was determined by X-ray diffraction measurements using Cu K$\alpha$ radiation (the Bruker D8-advance model). The phase purity and stoichiometry of the single crystals were confirmed by energy dispersive X-ray spectroscopy (EDX). The actual Ru:Cr ratio was evaluated by scanning a dozen spots of 50 μm size (Supplementary Fig. 1). The standard deviation from the mean value is evaluated to be -1 mol% Cr for all crystals. We measured dc magnetic susceptibility and magnetization with a SQUID (Quantum Design MPMS) and Physical Property Measurements System (Quantum Design PPMS Dynacool) for $B//ab$ and $B//c$ in the temperature range $T = 2$–300 K. High-field magnetization measurements were conducted at the Dresden High Magnetic Field Laboratory with a pulsed-field magnet (25 ms duration) using an induction method with a pickup coil device at $T = 2$ K. Specific heat experiments were carried out under applied fields of $B//c = 0$, 0.5, and 3 T in the temperature range of $T = 2$–200 K with a thermal relaxation method using a commercial set-up of Physical Property Measurements System.

The magnetic specific heat of $\alpha$-Ru$_{1-x}$Cr$_x$Cl$_3$ was obtained by subtracting the specific heat of the isostructural nonmagnetic counterpart ScCl$_3$. Using the Bouvier method[38], we scaled the specific heat data of ScCl$_3$ by the molecular mass and Debye temperature and then used this scaled specific heat data to evaluate the magnetic specific heat of the Cr-doped RuCl$_3$. In doing that, we assumed that the Debye temperature does not vary significantly with the small Cr concentration (Supplementary Figs. 11 and 12). The magnetic specific heat was fitted using a sum of two phenomenological functions[39], $S_m = \sum_{i=1,3} S_{m_i} = \sum_{i=1,3} \frac{\rho_i/2}{1 + \exp\left[\left(\frac{\beta_i + \gamma_i T_i/T}{1 + T_i/T}\right)\ln\frac{T_i}{T}\right]}$. Here, $\rho_i$ is the weighting factor with a scaled constraint of $\rho_1 + \rho_2 + \rho_3 = 2.14$ and $T_i$ is the crossover temperature. $\beta_i$ and $\gamma_i$ are the power exponents at high and low temperatures, respectively. The fitting parameters are evaluated to be $\rho_1 = 0.32(1)$, $\beta_1 = 2.9(2)$, $\gamma_1 = 5.18(9)$, $T_1 = 10.7(3)$ K, $\rho_2 = 0.41(5)$, $\beta_2 = 5.13(7)$, $\gamma_2 = 1.6(3)$, $T_2 = 24(4)$ K, $\rho_3 = 1.41(3)$, $\beta_3 = 3.88(9)$, $\gamma_3 = 0.7(2)$, and $T_3 = 70(7)$ K.

### Raman scattering

Raman scattering experiments were conducted in backscattering geometry with the excitation line $\lambda = 532$ nm of the DPSS SLM laser. The Raman scattering spectra were collected using a micro-Raman spectrometer (XperRam200VN, NanoBase) equipped with an air-cooled charge-coupled device (Andor iVac Camera). We employed a notch filter to reject Rayleigh scattering at low frequencies below 15 cm$^{-1}$. The laser beam with $P = 80$ μW was focused on a few-micrometer-diameter spot on the surface of the crystals using a ×40 magnification microscope objectives. The samples were mounted onto a $^4$He continuous flow cryostat by varying a temperature $T = 4$–300 K.

Phonon excitations below 200 cm$^{-1}$ were fitted using an asymmetric Fano profile $I(\omega) = I_0 \frac{(q+\epsilon)^2}{(1+\epsilon^2)}$, where $\epsilon = (\omega - \omega_0)/\Gamma$ and $\Gamma$ is the full width at half maximum (FWHM) in case of strong coupling between spin and lattice degree of freedom. $1/|q|$ provides a measure of the coupling strength between a magnetic continuum and optical phonons or conveys information about Majorana excitations.

### Muon spin relaxation/rotation

Muon spin-relaxation/rotation ($\mu$SR) measurements were conducted on the GPS[40] and the HAL-9500 spectrometers at the Paul Scherrer Institute (Villigen, Switzerland). For the GPS spectrometer measurements, a mosaic of $a$-axis coaligned single crystals (-0.5 g) was packed in an aluminum foil and attached to a sample holder. The Veto mode was activated to minimize the background signal. ZF- and TF-$\mu$SR experiments on the GPS spectrometer were performed in the spin-rotated mode, where the initial muon spins were rotated by 45° from the muon momentum direction ($c$-axis). It should be noted that $\alpha$-RuCl$_3$ shows anisotropic 2D XY-like magnetism, resulting in weaker spin correlations along the $c$-axis compared to those in the $ab$-plane. This makes it difficult to detect changes in the muon spin relaxation when the muon spins are directed along the $c$-axis. To minimize the contribution of spin correlations along the $c$-axis, up and down detectors were utilized in this spin-rotated mode. On the other hand, LF-$\mu$SR measurements on the GPS spectrometer were carried out in the longitudinal mode, where the initial muon spins were parallel to the $c$-axis. For the HAL-9500 experiments, a single piece of large single crystal ($8 \times 8 \times 1$ mm$^3$, -150 mg) was wrapped with a Ag foil and attached to a silver sample holder using GE varnish. All the measurements were carried out in the spin-rotated mode that the initial muon spins were rotated by 90° and lie in the $ab$-plane. The transverse fields ($B = 0$–3 T) were applied along the $c$-axis.

All obtained $\mu$SR spectra were analyzed with the software package MUSRFIT with GPU acceleration support[41–44]. The weak transverse-field (wTF) $\mu$SR spectra were fitted with a sum of an exponentially decaying cosine and a simple exponential function, $P_z(t) = f \cos(2\pi\nu_s t + \phi_s) \exp(-\lambda_s t) + (1-f) \exp(-\lambda_f t)$, where $f$ is the slow relaxing fraction, $\nu_s$ is the muon spin precession frequency, $\phi_s$ is a phase, and $\lambda_s$ ($\lambda_f$) is the muon spin-relaxation rate for the slow (fast) decaying component.

The zero-field (ZF) $\mu$SR data were well described by a sum of the Gaussian-broadened Gaussian (GbG) function with a simple exponential decay and a simple exponential function,

$$P_z(t) = f P_{\mathrm{GbG}}(t; \Delta_0, W) \exp(-\lambda_s t) + (1-f) \exp(-\lambda_f t)$$

The GbG depolarization function is defined as a convolution of the Gaussian Kubo-Toyabe function, characterizing a broader field distribution than the Gaussian field distribution,

$$P_{\mathrm{GbG}}(t) = a + (1-a)\left(\frac{1}{1+R^2\Delta_0^2 t^2}\right)^{3/2} \left(1 - \frac{\Delta_0^2 t^2}{1+R^2\Delta_0^2 t^2}\right) \exp\left[-\frac{\Delta_0^2 t^2}{2(1+R^2\Delta_0^2 t^2)}\right].$$

Here, $a$ is the tail fraction, $1-a$ is the damped relaxing fraction, $\Delta_0$ is the mean value, $W$ is the Gaussian width, and $R$ ($=W/\Delta_0$) is the relative Gaussian width of the Gaussian distribution, respectively. The GbG function well accounts for inhomogeneous static magnetic moments with short-range correlations[45–48]. Note that the ZF-$\mu$SR results of the nonmagnetic Ir$^{3+}$($J_{\mathrm{eff}} = 0$) substituted $\alpha$-Ru$_{1-x}$Ir$_x$Cl$_3$ are also well described by the identical model, suggesting the similar effects of magnetic (Cr$^{3+}$; $S = 3/2$) and nonmagnetic impurities on the Kitaev quantum spin system $\alpha$-RuCl$_3$[48].

The longitudinal-field (LF) $\mu$SR data were fitted by a sum of the static and the dynamic Gaussian Kubo-Toyabe functions in longitudinal fields,

$$P_z(t) = f P_{\mathrm{SGKT}}(t, \Delta_s, B_{\mathrm{LF}}) + (1-f) P_{\mathrm{DGKT}}(t, \Delta_f, \Gamma_f, B_{\mathrm{LF}}),$$

where, $P_{SGKT}$ ($P_{DGKT}$) are the dynamic (static) Gaussian Kubo-Toyabe function, $\Gamma_f$ is the local field fluctuation rate, $B_{LF}$ is the applied LF, and $\Delta_f$ ($\Delta_s$) is the local-field width at the muon interstitial sites. The internal field is evaluated to be $<B_{loc}>$ ~16.88 mT (Supplementary Fig. 16).

High transverse-field (hTF) $\mu$SR results were analyzed by the single histogram fit method. The positron histogram of the $i$-th detector $N_i(t)$ is given by $N_i(t) = N_{0,i}e^{-t/\tau_\mu}[1 + A_{0,i}P_i(t)] + N_{bkg,i}$, where $N_{0,i}$ is the total muon decay events at $t = 0$, $\tau_\mu$ is the mean lifetime of the muon (~2.2 μs), $A_{0,i}$ is the intrinsic asymmetry of the $i$-th detector, $P_i(t)$ is the time-dependent muon spin polarization, and $N_{bkg,i}$ is background events. We employed a sum of two Gaussian damped cosines for fittings, $P_i(t) = f\cos(2\pi\nu_s t + \phi_s)\exp[-\lambda_s t] + (1 - f)\cos(2\pi\nu_f t + \phi_f)\exp[-\lambda_f t]$, where $f$ is the relaxing fraction.

In general, to calculate the Knight shift, the narrow peak arising from the Ag sample holder is used as an internal reference. However, as shown in Fig. 4, the FFT spectra of $\alpha$-Ru$_{1-x}$Cr$_x$Cl$_3$ ($x = 0.04$) display the overlap of the background and the intrinsic sample signals at slightly higher than the applied field $B_{ext}$. Therefore, we used the peak position of the sharp signal at $T = 30$ K that was obtained from the analysis as the reference field for evaluating the Knight shift.

## Data availability

The magnetic susceptibility, specific heat, and Raman data generated in this study are provided in the Supplementary Information/Source Data file. The $\mu$SR data used in this study are available in the PSI database [http://musruser.psi.ch/cgi-bin/SearchDB.cgi]. Source data are provided with this paper.

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

## Acknowledgements

We gratefully acknowledge R. Scheuermann for assistance with the experiment and critical reading of our manuscript. The work at CAU and SKKU was supported by the National Research Foundation (NRF) of Korea (Grant no. RS-2023-00209121, 2020R1A5A1016518). A portion of this work was performed at the National High Magnetic Field Laboratory, which is supported by the National Science Foundation Cooperative Agreement No. DMR-1644779 and the State of Florida and the U.S. Department of Energy. This material is based upon work supported by the U.S. Department of Energy, Office of Science, National Quantum Information Science Research Centers. This work was supported by the Rare Isotope Science Project of the Institute for Basic Science funded by the Ministry of Science and ICT and NRF of Korea (2013M7A1A1075764). The research of M.V. was supported by the Deutsche Forschungsgemeinschaft (DFG) via the Cluster of Excellence ct.qmat (EXC 2147, project id 390858490) and via SFB 1143 (project id 247310070). Z.G. acknowledges support from the Swiss National Science Foundation (SNSF) through SNSF Starting Grant (No. TMSGI2_211750). The research of Y.C. was supported by the SungKyunKwan University and the BK21 FOUR (Graduate School Innovation) funded by the Ministry of Education (MOE, Korea) and NRF of Korea.

## Author contributions

K.-Y.C. designed and conceived the project. K.-Y.C. and S.L. planned the experiments. Y.S.C., S.-H.D., and C.H.L. synthesized the samples and characterized structural properties. S.L., Y.S.C., S.-H.D., and M.L. conducted magnetic property measurements. Y.S.C. performed the Raman scattering experiments and analyzed the Raman data. S.L., C.W., H.L., and Z.G. carried out μSR experiments, and S.L. and W.L. analyzed the data. Data analysis and figure preparation were performed by S.L., M.V., and K.-Y.C. The manuscript was written through the contributions of all authors.

## Competing interests

The authors declare no competing interests.
