## [Peer Review File · Nature Communications]

REVIEWER COMMENTS

Reviewer #1 (Remarks to the Author):

In this paper, the authors report their study of Kondo screening in Cr doped RuCl_3 , a QSL candidate material. The authors look at magnetic specific heat, susceptibility, muon Knight shift and spin relaxation rates to study possible signatures of Kondo screening arising from dispersing Majorana fermionic excitations in the extended Kitaev model related material RuCl_3 . The authors have made a detailed experimental study and the results are very interesting since people have been searching for possible signatures of QSLs in Kitaev model like materials. The overall findings and analysis of the manuscript are relevant and apt for publication in the journal. I have the below comments and questions that I believe should be addressed.

Since the work discusses results for the extended Kitaev's model and refers to the coupling constants of the model in various places, it will be good to add the Hamiltonian in the manuscript.

In lines 47-49, the authors mention the Kondo screening due to gauge flux binding but as far as I understand, the Kondo screening discussed in some of the references 8-11 is also the screening due to dispersive Majorana fermions and is the same as the metallic screening being proposed in the current manuscript. The local flux binding happens accompanying this screening.

The Ru-Cr interaction is mentioned to be of Heisenberg type in various places in the manuscript. It is not clear what is the reason for expecting a Heisenberg type interaction instead of a Kitaev like interaction as would be expected from the Cl environment.

In lines 108-109, the authors mention singlet vacancy and triplet vacancy. The references 9, 10 discuss single impurity and a pair of vacancies. There is no singlet or triplet formation happening, so the use of words singlet and triplet can be confusing.

Reviewer #2 (Remarks to the Author):

The manuscript reports mainly magnetisation, heat capacity and μSR measurements on Cr substituted $\alpha\text{-RuCl}_3$. The undoped compound undergoes long-range order below about 8 K and the ordering temperature is only weakly affected by Cr substitution. Essentially from their data above the ordering temperature, the authors claim the observation of metallic-like Kondo screening

of the dilute magnetic impurities. (A side note I would like to make is that 5-10% substitution in a honeycomb lattice can hardly be called dilute.)

I do not think that their claims are justified. I give below my comments.

Heat capacity: How to estimate the lattice heat capacity of doped RuCl₃ from that of non-magnetic ScCl₃? One can estimate the ratio of the Debye temperatures of the two compounds using Bouvier scaling [M. Bouvier, P. Lethuillier, D. Schmitt Phys. Rev., 43 (1991), p. 13137]. Nominally, the Debye temperature depends on the sound velocity which is proportional to the inverse square root of the density, i.e., $\sqrt{V/M}$. Further the TEMPERATURE axis has to be scaled by the Debye temperature and then at high temperature the data for ScCl₃ and doped RuCl₃ may be expected to coincide. (Do they?) Then, one may subtract the data for ScCl₃ from that of RuCl₃ to obtain the magnetic contribution. In the 25-50 K range, the lattice heat capacity is about 90% of the total heat capacity. The authors should put error bars on the inferred magnetic heat capacity.

The linear behaviour seen for the magnetic heat capacity has a non-zero intercept (C_m/T vs T is nowhere constant). Why? What happens for other Cr substitutions? Also, the system orders magnetically. So there will be a contribution to the specific heat from this as well.

μ SR and magnetisation: The logarithmic behaviours (which were seen in a narrow temperature range) are suggested to arise from magnetic impurities in a Kitaev Spin Liquid. But the given compound orders magnetically. So there will be a contribution to the susceptibility from that part also. How does one extract only the KSL contribution to the susceptibility/relaxation etc.?

Based on the above, I feel that the manuscript is not suitable for publication in Nature Communications.

Reviewer #3 (Remarks to the Author):

This manuscript presents extensive characterization data for a new doping series, α -Ru_{1-x}Cr_xCl₃, the end member of which is the notable material α -RuCl₃. This latter compound has been the subject of intense study in recent years due to a proximate spin liquid phase predicted by theory and seemingly confirmed by reports of deconfined spinon excitations and other unusual material properties. Indeed, the authors discuss the results of the current paper in the context of Majorana fermions and gauge fluxes, language which is adopted directly from the theory of Kitaev spin liquids and reflects their overall focus. The stated goal of studying the (Ru,Cr)Cl₃ doping series is to examine

the screening of Kondo impurities embedded in the putative spin liquid host material, with a presumed larger goal of learning more about the spin liquid phase.

The material α -RuCl₃ has a hexagonal sublattice and is known to order in a zig-zag pattern with a transition temperature of $T_N=6.5\text{K}$. Doping out the magnetic Ru³⁺ atoms with 3d Cr³⁺ has the potential to modify the material properties of via several different mechanisms: changing atom size, bonding lengths, electron density, itineracy, and the size of the spins, orbital moments and strength of spin-orbit coupling at the cation site. Accordingly, the authors attacked this complex and multi-faceted problem via several distinct probes of condensed matter, including x-ray diffraction (XRD), energy dispersive x-ray spectrometry (EDS), Raman scattering, magnetization, heat capacity and muon spin rotation/relaxation (μSR). Data from each of these techniques are considered in turn. The main conclusions are that (1) doping with Cr creates a small variation in ordering temperature and (2) the system remains metallic throughout the phase diagram, while (3) there are indications of logarithmic temperature divergence from various techniques which the authors associate with Kondo-like screening of the dopant moments. This last conclusion is considered significant in light of predictions of a Kitaev Kondo effect" and is held up by the authors as the central result of the work.

The development of doping phase diagrams is a powerful tool in the study of emergent phases of matter and in this context, the series α -Ru_{1-x}Cr_xCl₃ has some intrinsic interest. The single crystal samples seem to be well characterized, probes well-chosen, and both samples and data of highest quality. The functions used to fit data are mostly appropriate and conclusions well-justified, though I do have some concerns about fit ranges and assumptions inherent in some of the modelling; I detail these concerns in my comments below. The central observation of Kondo screening in this 4d material is interesting, though it is not yet clear to me how this observation alone furthers understanding of the spin liquid phase which underlies interest in these materials. I think the manuscript would benefit tremendously from the inclusion of additional text in Introduction and Discussion sections which discusses specific predictions for the Kitaev Kondo effect and exactly how measurement of this phenomena furthers understanding of the Kitaev materials. (i.e. Please tell me how observation of Kondo physics here is important, and if and how it is different from the traditional Kondo effect.) A detailed discussion on this point is of utmost importance and in my opinion entirely determines the level impact this work will have in the field.

Overall, I do feel this study deserves to be published in some form. Before doing so though, I would suggest the authors address the following questions and comments:

1.) Despite the importance of Kondo physics to the central conclusions of the paper, the Kondo effect itself is never explicitly defined. This seems like a major oversight which limits accessibility to the general reader.

2.) On the same note, the authors should explain how the Kondo effect "unravels the underlying gauge and topological character of a quantum spin liquid", as they say in the first line of their paper. How does the "Kitaev Kondo effect" manifest differently than the Kondo effect in, say, an f-electron metal with no particular topological character? Second quantization language aside, would one not expect Kondo impurities to be screened in correlated metal with no topological significance? What is

new here? Specific predictions for the material of interest would be particularly useful at this point in the paper to set the stage for future discussion.

3.) XRD: The methods section of the manuscript mentioned that the structure of the doping series was measured with powder diffraction, though the results of these measurements are never presented in either the main text or supplementary materials. XRD is better suited to comment on whether the Cr³⁺ dopants affect structure, which currently is one of the main conclusions of the Raman scattering section. It would be nice to compare the conclusions of the powder XRD study and the single crystal Raman study, when discussing structure.

4.) Magnetization, Curie-Weiss: Though the most exciting conclusions of magnetization analysis focus on the signatures of Kondo screening at low temperature, the manuscript does present the results of Curie-Weiss fits of high temperature data and spends a good portion of a paragraph discussing the results of these fits. As Curie-Weiss fits are notoriously sensitive to fit range, it was disappointing to see that neither the main text nor supplementary materials mentioned the temperature range which was fit to determine Curie and Weiss parameters. The closest thing was Supplementary figure 4a, which showed a semi-log plot of susceptibility for the single $x=0.04$ and a black line showing the C-W fit. Unfortunately, the only thing that I concluded from this plot was that the susceptibility for this material WAS NOT CURIE-WEISS LIKE for any significant region temperature! The Curie function is a high temperature expansion, only valid when all the S_z levels of a particular multiplet are appreciably populated. Based on Suppl Fig. 4a, the susceptibility has non-polynomial temperature dependence across the entire range investigated, and the authors seemingly chose a small intermediate range which they could approximate with a $1/T$ temperature dependence and used this range for all their fits. This is not acceptable, and the results of these fits are meaningless! These Curie fits are not central to the conclusions of the paper and so I strongly suggest dropping them. If the authors would like to keep Curie fits in the paper, they should repeat them using more defensible methodology. They should fit the highest temperatures available, from the highest measured down to where function deviations from the $1/(T-\theta)$. Any exclusion of high temperature data should be explicitly justified.

Fits should also include a temperature independent background term, reflecting Larmor and Pauli susceptibility terms which are significant for 4d metals and may explain the non-Curie-like temperature dependence. The temperature range fitted should be explicitly stated in the paper. I would also suggest showing data for all the crystals in the Supplementary materials to demonstrate how well the C-W function describes the data.

5.) Magnetization, Kondo: The discussion of logarithmic divergence in this section is closer to the theoretical predictions that I was looking for earlier in the paper and mentioned above. I was somewhat confused by the conversation though. Several different temperature dependences for different versions of Kondo screening were presented. Without looking at or referencing the data,

which version do the authors feel is justified by the doping of $S=3/2$ moments into a sea of Majorana fermions? Ultimately, the authors chose to fit to a functional form which was associated with a model for magnetic vacancies. Was this expected? If so, why? This is the central observation of the paper. The authors need to do some work explaining what they have learned when they observe temperature dependences consistent with a vacancy model after doping large $S=3/2$ moments. I am missing a connection here.

6.) Kondo continued, fit range: Similar to the Curie-Weiss fits, the authors seemed to pick out a finite temperature range that can be described by a particular Kondo screening model and claimed success, when in fact MOST of the temperature range does not follow the model identified. In this case, it is temperatures directly above the Neel temperature to temperature of $\sim 20\text{K}$. How did they choose the upper temperature of 20K for their fits? Over what temperature range is the model they chose expected to work? Is the $\sim 15\text{K}$ range where it did work consistent with the limitations of the chosen model?

7.) Heat capacity: The authors lay out an empirical description of their heat capacity, where different regions were dominated by different temperature functions which they associate with different physics. They claim "crossover temperatures" between the different regions, but then cite the temperature to three digits of precision with uncertainties. How did they mathematically determine the cross-over temperatures and do they really feel they know these (ambiguously defined) values to such high levels of precision?

8.) MuSR, muon spin direction: In the methods section, the authors claim that initial muon spins were rotated 45 degrees from the muon momentum direction (crystal c -axis). Was this an error? If so, both fields in the ab -plane and in the c -axis direction would be an admixture of longitudinal and transverse field configurations. This makes no sense. The same paragraph also says that the initial spin was rotated by 90 degrees, which is more conventional (but contradicts the previous statement). Please clarify.

9.) MuSR, heterogeneous magnetism: Both the chosen functions for zero-field and weak-transverse field fits imply that the magnetism in this system is heterogeneous: separated into ordered and disordered volumes. Why was this not mentioned in the main text?! Is it understood? Does the heterogeneity affect the predictions for the Kondo screening at all? How does the disordered volume scale with Cr-content?

This observation potentially has major implications for the conclusions of the paper and should be discussed more thoroughly. The authors should AT LEAST show WTF and ZF muon data for each sample investigated and show how the ordered fraction varies with doping. This system is supposed

to be a solid solution. If the ordered fraction varies linear with doping, the authors should consider the possibility that the system is self-separating into volumes of RuCl₃ and CrCl₃.

10.) MuSR, heterogeneous, part 2: In a traditional ZF-MuSR measurement of an ordered state (with 100% ordered volume), the non-oscillating fraction of the spectrum is associated with the fraction of the mean muon spin which is parallel to the local vector field direction at the muon site. In a polycrystal, the non-oscillating fraction is always 1/3, but this can vary significantly in a single crystal depending on the direction of the hyperfine field which arises below the ordering transition. Therefore, if the oscillating fraction f changes with doping, this might be a reflection of a changing ordered moment direction. The authors already have evidence for this from magnetization and should definitely explore this issue further.

11.) MuSR, ZF fit function: In writing the last comment, I noticed the authors fit their ZF spectra using a Gaussian-broadened Gaussian function, which they can see assumes a 2/3:1/3 ratio for oscillating and non-oscillating fractions in the spectra. As per my last comment, the 2/3:1/3 ratio is only appropriate for polycrystalline samples. They should relax this ratio for fits of ZF spectra of single crystals. They may find their samples are more homogeneous than they had been assuming.

12.) Discussion and conclusions: I mostly like the discussion, as it gives the kind of insight into the nature of the Kondo screening that I felt was missing in the previous sections. It would be nice if there was some discussion of different types of Kondo screening near the beginning of the paper, to allow the reader to appreciate the significance of the individual data sets as they were presented.

My biggest issue with this section is the ultimate paragraph, when the authors say “our findings raise the prospect of braiding impurity fluxes toward the implementation of quantum computation and realizing topological superconductivity through Kondo coupling of Kitaev QSL to conduction electrons”. I felt this sentence was completely unjustified. It is a powerful way to end the paper, but the authors need to draw a stronger line from their first observation of Kondo screening in doped RuCl₃ to braiding impurity fluxes. What do you mean here??

In summary, I think this manuscript represents a nice body of work on interesting materials with several nice conclusions. I brought up several items of concern involving fundamental analysis and interpretation, which should be addressed before moving forward with publication. However, I do think the authors can address all of the comments with the data they have, and I believe their conclusions will be stronger if the analysis is done properly. If these issues can be addressed, I do think this work is worthy of publication in Nature Communications.

Reviewer #1 (Remarks to the Author):

In this paper, the authors report their study of Kondo screening in Cr doped RuCl_3 , a QSL candidate material. The authors look at magnetic specific heat, susceptibility, muon Knight shift and spin relaxation rates to study possible signatures of Kondo screening arising from dispersing Majorana fermionic excitations in the extended Kitaev model related material RuCl_3 . The authors have made a detailed experimental study and the results are very interesting since people have been searching for possible signatures of QSLs in Kitaev model like materials. The overall findings and analysis of the manuscript are relevant and apt for publication in the journal. I have the below comments and questions that I believe should be addressed.

Authors) We would like to express our gratitude to the Reviewer for taking his/her time to read our manuscript and providing valuable comments. We appreciate his/her positive evaluation of the significance and novelty of our work. In this response, we will address the issues/questions raised by the Reviewer in a point-by-point manner.

Since the work discusses results for the extended Kitaev's model and refers to the coupling constants of the model in various places, it will be good to add the Hamiltonian in the manuscript.

Authors) Following the Reviewer's suggestion, we have added the spin Hamiltonian explicitly in the manuscript to improve its readability.

In lines 47-49, the authors mention the Kondo screening due to gauge flux binding but as far as I understand, the Kondo screening discussed in some of the references 8-11 is also the screening due to dispersive Majorana fermions and is the same as the metallic screening being proposed in the current manuscript. The local flux binding happens accompanying this screening.

Authors) The Reviewer is correct that, in the theory of Refs [10,11], the impurity moment is screened by mobile Majorana fermions. However, in the low-temperature limit investigated in these papers, the Majorana fermions of the Kitaev model are semimetallic and can by themselves not screen the impurity. Screening is only facilitated by binding a gauge flux which induces metallic behavior near the impurity and thus enables screening. The present experimental situation differs in that the metallic behavior of Majorana fermions occurs at elevated temperatures "automatically" due to the global presence of thermally excited gauge fluxes; hence no explicit binding of fluxes to impurities is required. We have clarified this point in the text.

The Ru-Cr interaction is mentioned to be of Heisenberg type in various places in the manuscript. It is not clear what is the reason for expecting a Heisenberg type interaction instead of a Kitaev like interaction as would be expected from the Cl environment.

Authors) We agree with the Reviewer that a proper description of the Ru-Cr interaction is important. Accordingly, we have added sentences to explain the dominant Heisenberg character of the Ru-Cr interaction.

Before delving into theoretical reasoning, we recall the magnetic susceptibility data presented in Fig. 2 of the main text. It is evident from these data that the XY-like anisotropy, as quantified by χ_{ac}/χ_c , systematically decreases with increasing Cr content. This trend alludes to the growth of isotropic Heisenberg interaction with x.

The Kitaev interaction and other anisotropic interactions (the so-called Gamma-terms) between Ir^{4+} or Ru^{3+} (d^5) ions originate from the formation of spin-orbit-entangled $j_{\text{eff}}=1/2$ local moments at metal cations, further assisted by the Hund's coupling between $j_{\text{eff}}=1/2$ and $3/2$ states [see Jackeli and Khaliullin, Phys. Rev. Lett. **102**, 017205 (2009) or Rau, Lee, and Kee, Phys. Rev. Lett. **112**, 077204 (2014)]. Because Cr^{3+} ions are in the high-spin d^3 $S=3/2$ configuration, they are orbitally inactive and are unable to provide multiple anisotropic and spin-dependent exchange paths, which is required for the presence of Kitaev and anisotropic exchange interactions. Edge-sharing geometry between nearest-neighboring Cl octahedra is one necessary condition for the realization of the Kitaev interaction, but without an active orbital degree of freedom and SOC it alone cannot generate Kitaev interactions.

In lines 108-109, the authors mention singlet vacancy and triplet vacancy. The references 9, 10 discuss single impurity and a pair of vacancies. There is no singlet or triplet formation happening, so the use of words singlet and triplet can be confusing.

Authors) We appreciate the Reviewer for pointing out the problematic terminology. As per the Reviewer's suggestion, we have removed the words "singlet vacancy" and "triplet vacancy" from the manuscript, and we have revised the discussion of the previously published theory results.

Reviewer #2 (Remarks to the Author):

The manuscript reports mainly magnetisation, heat capacity and μ SR measurements on Cr substituted α - RuCl_3 . The undoped compound undergoes long-range order below about 8 K and the ordering temperature is only weakly affected by Cr substitution. Essentially from their data above the ordering temperature, the authors claim the observation of metallic-like Kondo screening of the dilute magnetic impurities. (A side note I would like to make is that 5-10% substitution in a honeycomb lattice can hardly be called dilute.)

I do not think that their claims are justified. I give below my comments.

Authors) We thank the Reviewer for taking his/her time to read our manuscript and providing us with insightful comments. We have taken the Reviewer's comments into careful consideration and have addressed all the points raised in the revised version of the manuscript. We hope that our responses have improved the overall quality of the manuscript.

Heat capacity: How to estimate the lattice heat capacity of doped RuCl_3 from that of non-magnetic ScCl_3 ? One can estimate the ratio of the Debye temperatures of the two compounds using Bouvier scaling [M. Bouvier, P. Lethuillier, D. Schmitt Phys. Rev., 43 (1991), p. 13137]. Nominally, the Debye temperature depends on the sound velocity which is proportional to the inverse square root of the density, i.e., $\sqrt{V/M}$. Further the TEMPERATURE axis has to be scaled by the Debye temperature and then at high temperature the data for ScCl_3 and doped RuCl_3 may be expected to coincide. (Do they?) Then, one may subtract the data for ScCl_3 from that of RuCl_3 to obtain the magnetic contribution. In the 25-50 K range, the lattice heat capacity is about 90% of the total heat capacity. The authors should put error bars on the inferred magnetic heat capacity.

Authors) We appreciate the Reviewer for raising this issue. We are well aware of the so-called Bouvier method and have used it to scale the specific heat data of ScCl_3 in the same nanner as described in the literatures [Nat. Phys. **13**, 1079 (2017) and arXiv:2203.13407v1 (2022)]. Specifically, we scaled the specific heat data of ScCl_3 by the molecular mass and Debye temperature using the Bouvier method, and then used this scaled C_p data to evaluate the magnetic specific heat of the Cr-doped RuCl_3 . In doing that, we assumed that the Debye temperature does not vary significantly with the small Cr concentration. In the revised manuscript, we added the corresponding paragraphs in Methods to describe this procedure in more detail.

Furthermore, we have included error bars in the magnetic specific heat data (Fig. 3c). We have added error bars to the data in the revised manuscript to more accurately represent the uncertainty in the measurements.

The linear behaviour seen for the magnetic heat capacity has a non-zero intercept (C_m/T vs T is nowhere constant). Why? What happens for other Cr substitutions? Also, the system orders magnetically. So there will be a contribution to the specific heat from this as well.

Authors) We thank the Reviewer for bringing this to our attention. As the Reviewer may know, the magnetic specific heat in a Kitaev spin system bears thermal fractionalization of spins [see J. Nasu et al., Phys. Rev. B **92**, 115122 (2015)].

As shown in the adapted figure, the heat capacity of a perfect Kitaev model shows a two-peak structure, with the low- T peak arising from flux freezing and the high- T peak from the establishment of short-range spin correlation. The crossover between these two peaks induces a metallic-like regime for the itinerant Majoranas caused by thermally fluctuating fluxes. Whether or not C/T displays a visibly constant behavior in this temperature regime depends, e.g., on the magnetic anisotropy, as can be inferred from Figs 1 and 4 of Nasu et al.

In a real Kitaev system, such as α - RuCl_3 , the situation is significantly more complicated. At very low T long-range magnetic order may be present, which then masks the low- T behavior of the spin liquid and leads to a lambda-like anomaly around T_N . In the spin-liquid regime, i.e., above T_N , the dynamics of visons, not present in the pure Kitaev model, introduces an additional energy scale and additional contributions to the heat capacity which can extend to temperatures of the order of the couplings beyond Kitaev. Hence, a regime of purely linear $C(T)$ is unlikely to be expected. In interpreting our experimental data we take the large background in C/T below 100 K as evidence for metallic behavior of Majorana fermions.

To complete our dataset, we further measured the specific heat for the $x=0.02$ compound, represented in Supplementary Fig. 7 and the right panel. Overall, the trend falls between the $x=0$ and $x=0.04$ data (Fig. 3c in the main text). As with the $x=0$ and $x=0.04$ data, we see the emergence of a linear T contribution in C_m below 100 K (in addition to a constant background), indicating a Majorana metallic state above T_N .

muSR and magnetisation: The logarithmic behaviours (which were seen in a narrow temperature range) are suggested to arise from magnetic impurities in a Kitaev Spin Liquid. But the given compound orders magnetically. So there will be a contribution to the susceptibility from that part also. How does one extract only the KSL contribution to the susceptibility/relaxation etc.?

Authors) As the Reviewer rightly pointed out, the total magnetic susceptibility involves various contributions and crossovers. In response to the Reviewer's comments, we have reanalyzed the magnetic susceptibility above T_N using a more refined fit which includes both logarithmic behavior at lower T and a cross-over power-law $T^{-\alpha}$ at higher T . The results are presented in Supplementary Fig. 6. We could identify the crossover from power-law to logarithmic dependence around $T^* \sim 30$ K. We interpret this as follows: Above T^* , the susceptibility is dominated by a bulk-driven Curie-Weiss-type behavior which is cut off by the magnetic interactions while it displays a weaker logarithmic divergence below T^* driven by Kondo impurities. The latter is eventually cut-off by the onset of magnetic order. Since the order is antiferromagnetic in nature, we do not expect to see strong precursor effects in the uniform susceptibility. The comparison with the $x=0$ data in Supplementary Fig 3 confirms that the logarithmic increase between T_N and T^* is impurity-induced.

Further, we note that unlike the static magnetic susceptibility, dynamic probes such as muSR can detect an intrinsic magnetic susceptibility that is separated from impurity/defect contributions. Thus, the logarithmic behavior, lacking for the pristine sample, bears spin correlations engendered by the insertion of magnetic impurities. For the readers, we explicitly state this in the revised manuscript.

Based on the above, I feel that the manuscript is not suitable for publication in Nature Communications.

Authors) Based on these revisions and the recommendations of publication from the other two Reviewers, we hope Reviewer #2 finds that the updated manuscript is now suitable for publication in Nature Communications. The search for Kondo physics in quantum spin liquids has been a highly pursued topic. Although several theoretical scenarios have been proposed, its experimental identification is extremely challenging. In this regard, the observation of Kondo phenomena in a Kitaev-related system will advance our understanding of the underlying Kondo mechanism in quantum magnets.

Reviewer #3 (Remarks to the Author):

This manuscript presents extensive characterization data for a new doping series, $\alpha\text{-Ru}_{1-x}\text{Cr}_x\text{Cl}_3$, the end member of which is the notable material $\alpha\text{-RuCl}_3$. This latter compound has been the subject of intense study in recent years due to a proximate spin liquid phase predicted by theory and seemingly confirmed by reports of deconfined spinon excitations and other unusual material properties. Indeed, the authors discuss the results of the current paper in the context of Majorana fermions and gauge fluxes, language which is adopted directly from the theory of Kitaev spin liquids and reflects their overall focus. The stated goal of studying the $(\text{Ru,Cr})\text{Cl}_3$ doping series is to examine the screening of Kondo impurities embedded in the putative spin liquid host material, with a presumed larger goal of learning more about the spin liquid phase.

The material $\alpha\text{-RuCl}_3$ has a hexagonal sublattice and is known to order in a zig-zag pattern with a transition temperature of $T_N=6.5$ K. Doping out the magnetic Ru^{3+} atoms with 3d Cr^{3+} has the potential to modify the material properties via several different mechanisms: changing atom size, bonding lengths, electron density, itineracy, and the size of the spins, orbital moments and strength of spin-orbit coupling at the cation site. Accordingly, the authors attacked this complex and multi-faceted problem via several distinct probes of condensed matter, including x-ray diffraction (XRD), energy dispersive x-ray spectrometry (EDS), Raman scattering, magnetization, heat capacity and muon spin rotation/relaxation (μSR). Data from each of these techniques are considered in turn. The main conclusions are that (1) doping with Cr creates a small variation in ordering temperature and (2) the system remains metallic throughout the phase diagram, while (3) there are indications of logarithmic temperature divergence from various techniques which the authors associate with Kondo-like screening of the dopant moments. This last conclusion is considered significant in light of predictions of a "Kitaev Kondo effect" and is held up by the authors as the central result of the work.

The development of doping phase diagrams is a powerful tool in the study of emergent phases of matter and in this context, the series $\alpha\text{-Ru}_{1-x}\text{Cr}_x\text{Cl}_3$ has some intrinsic interest. The single crystal samples seem to be well characterized, probes well-chosen, and both samples and data of highest quality. The functions used to fit data are mostly appropriate and conclusions well-justified, though I do have some concerns about fit ranges and assumptions inherent in some of the modelling; I detail these concerns in my comments below. The central observation of Kondo screening in this 4d material is interesting, though it is not yet clear to me how this observation alone furthers understanding of the spin liquid phase which underlies interest in these materials. I think the manuscript would benefit tremendously from the inclusion of additional text in Introduction and Discussion sections which discusses specific predictions for the Kitaev Kondo effect and exactly how measurement of this phenomena furthers understanding of the Kitaev materials. (i.e. Please tell me how observation of Kondo physics here is important, and if and how it is different from the

traditional Kondo effect.) A detailed discussion on this point is of utmost importance and in my opinion entirely determines the level impact this work will have in the field.

Overall, I do feel this study deserves to be published in some form. Before doing so though, I would suggest the authors address the following questions and comments:

Authors) We appreciate the Reviewer for his/her meticulous reading of the manuscript and constructive comments, which helped us to improve the presentation style and to enhance the clarity of our results. In order to meet the Reviewer's concerns, we have revised our manuscript significantly and addressed all the issues. Please, find below our point-by-point responses to the Reviewer's issues/questions.

1.) Despite the importance of Kondo physics to the central conclusions of the paper, the Kondo effect itself is never explicitly defined. This seems like a major oversight which limits accessibility to the general reader.

Authors) We are grateful to the Reviewer for his/her suggestion to improve the comprehensibility of our work for the readers. Accordingly, we have included a concise definition of the Kondo effect in Introduction of our manuscript.

2.) On the same note, the authors should explain how the Kondo effect "unravels the underlying gauge and topological character of a quantum spin liquid", as they say in the first line of their paper. How does the "Kitaev Kondo effect" manifest differently than the Kondo effect in, say, an f-electron metal with no particular topological character? Second quantization language aside, would one not expect Kondo impurities to be screened in correlated metal with no topological significance? What is new here? Specific predictions for the material of interest would be particularly useful at this point in the paper to set the stage for future discussion.

Authors) We appreciate the Reviewer for bringing up this issue. While the Reviewer is correct in saying that Kondo screening also happens in a non-topological host metal, there are two key things here: In a generic insulating magnet, magnetic excitations are bosonic and do not lead to a Kondo effect. Only fractionalized spin liquids can feature fermionic excitations which, in some cases, can display a Fermi surface. Hence, the presence of logarithmic behavior, e.g. in $\chi(T)$, in a magnetic insulator is evidence for the fractionalized (and hence topological) character of the host. Moreover, in the pure Kitaev model, the itinerant fermions are semimetallic, leading to a very different Kondo effect. Therefore, the presence of logarithms also evidences metallic (instead of semimetallic) behavior, which then is most likely cause by a thermally fluctuating flux (or vison) background. We have clarified this in the revised manuscript.

3.) XRD: The methods section of the manuscript mentioned that the structure of the doping series was measured with powder diffraction, though the results of these measurements are never presented in either the main text or supplementary

materials. XRD is better suited to comment on whether the Cr³⁺ dopants affect structure, which currently is one of the main conclusions of the Raman scattering section. It would be nice to compare the conclusions of the powder XRD study and the single crystal Raman study, when discussing structure.

Authors) Following the Reviewer's recommendation, we have added XRD data in Supplementary Note and Figure, as presented below. Regrettably, we found that the old powder XRD data are no longer available. Instead, we present the single-crystal XRD patterns and the c-axis lattice parameter as a function of x. We find that the c-axis parameter increases quasilinearly with increasing Cr concentration.

4.) Magnetization, Curie-Weiss: Though the most exciting conclusions of magnetization analysis focus on the signatures of Kondo screening at low temperature, the manuscript does present the results of Curie-Weiss fits of high temperature data and spends a good portion of a paragraph discussing the results of these fits. As Curie-Weiss fits are notoriously sensitive to fit range, it was disappointing to see that neither the main text nor supplementary materials mentioned the temperature range which was fit to determine Curie and Weiss parameters. The closest thing was Supplementary figure 4a, which showed a semi-log plot of susceptibility for the single $x=0.04$ and a black line showing the C-W fit. Unfortunately, the only thing that I concluded from this plot was that the susceptibility for this material WAS NOT CURIE-WEISS LIKE for any significant region temperature! The Curie function is a high temperature expansion, only valid when all the S_z levels of a particular multiplet are appreciably populated. Based on Suppl Fig. 4a, the susceptibility has non-polynomial temperature dependence across the entire range investigated, and the authors seemingly chose a small intermediate range which they could approximate with a $1/T$ temp dependence and used this range for all their fits. This is not acceptable, and the results of these fits are meaningless! These Curie fits are not central to the conclusions of the paper and so I strongly suggest dropping them. If the authors would like to keep Curie fits in the paper, they should repeat them using more defensible methodology. They should fit the highest temperatures available, from the highest measured down to where function deviations from the $1/(t-\theta)$. Any exclusion of high temperature data should be explicitly justified.

Fits should also include a temperature independent background term, reflecting Larmor and Pauli susceptibility terms which are significant for 4d metals and may explain the non-Curie-like temperature dependence. The temperature ranged fitted should be explicitly stated in the paper. I would also suggest showing data for all the crystals in the Supplementary materials to demonstrate how well the C-W function describes the data.

Authors) We regret that the solid black line in Supplementary Fig. 6a, representing the logarithmic dependence of the magnetic susceptibility for $B//ab$, was misinterpreted as the Curie-Weiss fit. The solid lines in Figs. 3(a), (b) in the main text are also the logarithmic dependence. On the other hand, the dashed line in Fig. 3(a) is indeed the Curie-Weiss fit for $x=0.07$, although it may appear as a guide line to some readers because the temperature window is limited below 100 K. We stress that we analyzed the magnetic susceptibility using the logarithmic dependence $\ln(D/T)$ at low temperatures to address Kondo physics.

At any rate, to avoid any confusion, we have revised Fig. 3(a) and added a paragraph to specify the Curie-Weiss fittings. As evident from the T-linear dependence in the magnetic specific heat, a Kitaev paramagnetic state begins to appear at around 100 K. As such, while a thermally fluctuating paramagnetic lies above 100 K.

For the readers, we added a new figure for presenting the magnetic susceptibility data vs. the Curie-Weiss fitting results in Supplementary Note and Figure. As shown below, the Curie-Weiss fits (dashed lines) are valid for temperatures above 100-180 K.

5.) Magnetization, Kondo: The discussion of logarithmic divergence in this section is closer to the theoretical predictions that I was looking for earlier in the paper and

mentioned above. I was somewhat confused by the conversation though. Several different temperature dependences for different versions of Kondo screening were presented. Without looking at or referencing the data, which version do the authors feel is justified by the doping of $S=3/2$ moments into a sea of Majorana fermions? Ultimately, the authors chose to fit to a functional form which was associated with a model for magnetic vacancies. Was this expected? If so, why? This is the central observation of the paper. The authors need to do some work explaining what they have learned when they observe temperature dependences consistent with a vacancy model after doping large $S=3/2$ moments. I am missing a connection here.

Authors) We thank the Reviewer for pointing this out. The observed logarithmic dependence is consistent with a “conventional” Kondo effect between dilute magnetic impurities and metallic-like Majorana fermions. To date, there are no theoretical studies of $S=3/2$ impurities in a Kitaev spin liquid, neither in the low- T regime nor at elevated T . It is plausible that the relevant single-impurity model is that of a three-channel spin- $3/2$ Kondo model. Its asymptotic low- T physics can be expected to be similar to that of Ref. 11, where Kondo screening in the globally semimetallic environment is facilitated only by flux binding. At elevated T , we similarly expect “conventional” metallic screening, with log crossover signatures, due to thermally excited fluxes (or visons) rendering the Majorana dynamics metallic. We have revised the text accordingly.

6.) Kondo continued, fit range: Similar to the Curie-Weiss fits, the authors seemed to pick out a finite temperature range that can be described by a particular Kondo screening model and claimed success, when in fact MOST of the temperature range does not follow the model identified. In this case, it is temperatures directly above the Neel temperature to temperature of $\sim 20\text{K}$. How did they choose the upper temperature of 20K for their fits? Over what temperature range is the model they chose expected to work? Is the $\sim 15\text{K}$ range where it did work consistent with the limitations of the chosen model?

Authors) We agree with the Reviewer that this is a significant issue. By comparing the temperature dependence of both heat capacity (see Fig 3 and response to Referee 2) and susceptibility (see point 4 above) between the pristine and the doped samples, it becomes clear that the main effect of magnetic impurities is visible between T_N and $25\text{-}30\text{ K}$. Therefore, we have chosen essentially this range for the fitting.

7.) Heat capacity: The authors lay out an empirical description of their heat capacity, where different regions were dominated by different temperature functions which they associate with different physics. They claim “crossover temperatures” between the different regions, but then cite the temperature to three digits of precision with uncertainties. How did they mathematically determine the cross-over temperatures and do they really feel they know these (ambiguously defined) values to such high levels of precision?

Authors) The empirical function used in our analysis is the Schottky-like function proposed in the reference [Phys. Rev. B 93, 174425 (2016)]. The crossover temperature is determined by the Schottky-like peak in the magnetic specific heat, corresponding to the temperature where the exponentially decaying magnetic entropy has the mean value. Previous studies [Phys. Rev. B **93**, 174425 (2016) for Na₂IrO₃ and Nat. Phys. **13**, 1079 (2017) for RuCl₃] have shown that the fittings with the empirical function well reproduce the two-step release of the magnetic entropy with small uncertainties for β_i , γ_i , ρ_i , and the relatively large uncertainty for T_i . In our case, the uncertainty is comparable with the previous reports. Noticeably, the extracted crossover temperatures should be taken as a thermodynamic quantification of the thermal fractionalization process of spins.

In response to the Reviewer#2's request, we measured additional specific heat, which lead to the identification to another temperature scale below the onset $T \sim 100$ K temperature of spin fractionalization. To enhance the clarity, we have inserted error bars and revised the corresponding text.

8.) MuSR, muon spin direction: In the methods section, the authors claim that initial muon spins were rotated 45 degrees from the muon momentum direction (crystal c -axis). Was this an error? If so, both fields in the ab -plane and in the c -axis direction would be an admixture of longitudinal and transverse field configurations. This makes no sense. The same paragraph also says that the initial spin was rotated by 90 degrees, which is more conventional (but contradicts the previous statement). Please clarify.

Authors) For the ZF and weak TF experiments on the GPS spectrometer, the muon spins were rotated by 45 degrees from the c -axis. As it is well known, α -RuCl₃ shows anisotropic 2D XY-like magnetism. When the muon spins are directed along the c -axis in the ZF measurements, we can not observe the change of the muon spin relaxation for α -RuCl₃. This is due to the fact that the spin correlations along the c -axis are much weaker than those in the ab -plane, which makes it difficult to observe changes in the muon spin relaxation when the muon spins are directed along the c -axis. Therefore, for the ZF and weak TF- μ SR, we rotated the muon spin by 45 degrees from the c -axis to observe the in-plane spin correlations.

In addition, for the ZF and weak TF data, the up and down detectors were used to minimize the contribution of spin correlations along the c -axis. In contrast, for the LF experiments, the muon spins were parallel to the c -axis in order to decouple the muon spin from internal fields. Lastly, for the high TF experiments on the HAL-9500 spectrometer, the muon spins were in the ab -plane. Using this geometry, we can trace the development of in-plane spin correlations. To our knowledge, a spin-rotated mode (the rotation of muons by 45 degrees) is a routine procedure for studying spin correlations in single crystals.

To avoid any confusion to the readers, we have clearly described the experimental geometry used for ZF, LF, weak TF, and high TF measurements in Methods section of

the revised manuscript.

9.) MuSR, heterogeneous magnetism: Both the chosen functions for zero-field and weak-transverse field fits imply that the magnetism in this system is heterogeneous: separated into ordered and disordered volumes. Why was this not mentioned in the main text?! Is it understood? Does the heterogeneity affect the predictions for the Kondo screening at all? How does the disordered volume scale with Cr-content?

This observation potentially has major implications for the conclusions of the paper and should be discussed more thoroughly. The authors should AT LEAST show WTF and ZF muon data for each sample investigated and show how the ordered fraction varies with doping. This system is supposed to be a solid solution. If the ordered fraction varies linear with doping, the authors should consider the possibility that the system is self-separating into volumes of RuCl_3 and CrCl_3 .

Authors) We appreciate the Reviewer's insightful comments. Since a few percentage of $S=3/2$ Cr^{3+} ions replace $J=1/2$ Ru^{3+} , we naturally expect inhomogeneities on an atomic scale. It is well known that substituted Cr spins are randomly distributed, as supported by previous literature [Inorg. Chem. **58**, 6659 (2019), Phys. Rev. B **99**, 214410 (2019)] and our Raman and XRD measurements. We did not observe any structural domains or phase segregation based on the absence of phonon modes pertinent to CrCl_3 . We explicitly state this in the revised manuscript.

Unfortunately, we only performed ZF- and wTF- μ SR measurements on $\alpha\text{-Ru}_{1-x}\text{Cr}_x\text{Cl}_3$ ($x=0.04$) to study the evolution of a magnetic ground state. Therefore, it is not possible to systematically compare the ordered fraction as a function of the Cr concentration. Nonetheless, we recall that $\alpha\text{-Ru}_{1-x}\text{Ir}_x\text{Cl}_3$ shows a similar inhomogeneous magnetism as published in Phys. Rev. B **98**, 014407 (2018). In $\alpha\text{-Ru}_{1-x}\text{Ir}_x\text{Cl}_3$, the fraction of the Gaussian-broadened component was evaluated to be 0.74 (0.5) for $x=0.16$ (0.2). Thus, the ordered fraction is not linearly dependent on the Ir concentration.

Although we have not traced the evolution of a magnetic ground state using a microscopic probe, our magnetic susceptibility (see Supplementary Fig. 3) reveals that the effective magnetic moment and the Neel temperature slightly decrease with increasing x up to 0.03 and show little variation with x at $x=0.03$ -0.07. Besides, the magnetic continuum probed by Raman spectroscopy (see Fig. 1c of the main text) hardly varies with x in the studied x range. This means that a quantum-disordered state does not experience a notable change, yet the ordered magnetic moment will be drastically altered with increasing Cr content. At any rate, the x dependence of a magnetic ground state is a separate topic, requiring extensive investigation. We just stress that our main conclusion is drawn above the magnetically ordered temperature.

As to the effect of heterogenous magnetism on Kondo screening, the $S=3/2$ impurities imbedded in the background of a Majorana metallic state is an essential ingredient to observe the Kondo screening. Above T_N , the Cr^+ ions behave like paramagnetic ions and, thus, the inhomogeneity does not significantly affect Kondo physics, unlike the magnetically ordered state below T_N .

10.) MuSR, heterogeneous, part 2: In a tradition ZF-MuSR measurement of an ordered state (with 100% ordered volume), the non-oscillating fraction of the spectrum is associated with the fraction of the mean muon spin which is parallel to the local vector field direction at the muon site. In a polycrystal, the non-oscillating fraction is always 1/3, but this can vary significantly in a single crystal depending on the direction of the hyperfine field which arises below the ordering transition. Therefore, if the oscillating fraction f changes with doping, this might be a reflections of a changing ordered moment direction. The authors already have evidence for this from magnetization and should definitely explore this issue further.

Authors) We appreciate the Reviewer's comment. Unfortunately, we only conducted ZF- and wTF- μ SR measurements on α -Ru_{1-x}Cr_xCl₃ ($x=0.04$). Therefore, the change of the ordered fraction as a function of the Cr concentration can not be determined. The systematic x evolution of the magnetic moments is beyond the scope of the present work and can be an excellent subject for future work combined with neutron diffraction.

11.) MuSR, ZF fit function: In writing the last comment, I noticed he authors fit their ZF spectra using a Gaussian-broadened Gaussian function, which they can see assumes a 2/3:1/3 ratio for oscillating an non-oscillating fractions in the spectra. As per my last comment, the 2/3:1/3 ratio is only appropriate for polycrystalline samples. They should relax this ratio for fits of ZF spectra of single crystals. They may find their samples are more homogeneous than they had been assuming.

Authors) In principle, we agree with the Reviewer's comments. As the Reviewer pointed out, the 1/3 and 2/3 components originated from the randomly distributed local fields. However, for α -Ru_{1-x}Cr_xCl₃, the Cr substitution effectively introduces the random fields into the host system through exchange randomness and bond disorder induced by the Cr impurities. Given that the magnetic response is dominated by the SU(2) randomly distributed impurities, the 1/3 and 2/3 components in the Gaussian-broadened Gaussian function can be justified even for single crystals.

12.) Discussion and conclusions: I mostly like the discussion, as it gives the kind of insight into the nature of the Kondo screening that I felt was missing in the previous sections. It would be nice if there was some discussion of different types of Kondo screening near the beginning of the paper, to allow the reader to appreciate the significance of the individual data sets as they were presented.

My biggest issue with this section is the ultimate paragraph, when the authors say "our findings raise the prospect of braiding impurity fluxes toward the implementation of quantum computation and realizing topological superconductivity through Kondo coupling of Kitaev QSL to conduction electrons". I felt this sentence was completely unjustified. It is a powerful way to end the paper, but the authors need to draw a stronger line from their first observation of Kondo screening in doped RuCl₃ to braiding impurity fluxes. What do you mean here??

Authors) The Reviewer is correct that this may be too far fetched. The statement has two parts, both of which draw from having a realization of a Kitaev-like spin liquid in RuCl₃. In a Kitaev spin liquid at low T, certain impurities can bind fluxes as well as Majoranas [9,11], and the ability to manipulate impurities (e.g. by STM techniques) would pave the way to their braiding. Coupling a Kitaev spin liquid to conduction electrons can produce a topological superconductor [37,38]. As the latter issue is not directly related to the present Cr doping study, we have changed the concluding paragraph.

In summary, I think this manuscript represents a nice body of work on interesting materials with several nice conclusions. I brought up several items of concern involving fundamental analysis and interpretation, which should be addressed before moving forward with publication. However, I do think the authors can address all of the comments with the data they have, and I believe their conclusions will be stronger if the analysis is done properly. If these issues can be addressed, I do think this work is worthy of publication in Nature Communications.

REVIEWER COMMENTS

Reviewer #1 (Remarks to the Author):

Dear Editor,

The authors have addressed the comments satisfactorily. I am happy to recommend publication of the article in the journal.

Best regards.

Reviewer #4 (Remarks to the Author):

The manuscript claims the observation of a metallic-like spin-liquid Kondo effect of $S = 3/2$ Cr^{3+} impurities embedded in a host $S = 1/2$ Kitaev QSL candidate material, $\alpha\text{-RuCl}_3$, at intermediate temperatures. This is based on the results of an impressive range of experimental techniques, including magnetisation, specific heat, muon spectroscopy, Raman spectroscopy and X-ray diffraction. Overall, I feel that the results are interesting, (for the most part) well-analysed, and that they deserve publication in the journal, as long as the authors properly address the questions that have arisen.

I have been asked by the editor to look over the authors' responses to the comments and questions made by Reviewers #2 and #3 who were not available to assess these themselves. In my opinion, the authors' have appropriately addressed all concerns of Reviewer #2 and most concerns of Reviewer #3.

Below is my assessment of the outstanding concerns of Reviewer #3 that have, in my opinion, not yet been resolved fully:

3.) XRD and 9.) MuSR, heterogeneous magnetism:

The authors claim that the original powder XRD data are no longer available, and they present new single-crystal XRD results in their stead. However, in the revised manuscript, they still claim that powder X-ray diffraction was used to determine the crystal structure under “Structural and thermodynamic measurements” part of the methods section, and Supplementary Note 1 still has “powder XRD” in its title.

Frankly, I find this unacceptable. If the authors can not show the powder XRD data, all mentions of it should be removed from the manuscript. Leaving such mentions in even contradicts their standard data availability claim that “data ... are available from the corresponding author upon reasonable request”. Luckily, it seems that no other results in the manuscript pertain to powder samples, as the rest deals with single crystals and single-crystal XRD is presented, so removing mentions of powder XRD results should not in itself invalidate the main claims of the manuscript.

However, the way that the single-crystal XRD data that replaces it are presented in Supplementary Fig. 2 is problematic. Even though the data is claimed to have been refined using FullProf, and a fitted lattice parameter is presented, the actual fits and their residuals are not shown, and standard quality-of-fit measures like R-factors and χ^2 are not mentioned. Furthermore, the plots are rasterized in a rather low resolution, making it hard to visually discern peak shapes to confirm that no peak splitting, which could point toward structural changes or phase separation, are present.

This is especially problematic in light of Reviewer #3’s question 9.) about the possibility of self-separation of the system into larger volumes of RuCl₃ and CrCl₃ (which would be consistent with muon spectroscopy results), where, in response, the authors claim that XRD data helped them determine that no phase segregation occurs in their substituted samples. To substantiate this claim the authors should present their full single-crystal XRD fits, residuals, and quality-of-fit measures for the model of single-phase α -Ru_{1-x}Cr_xCl₃. These should furthermore be compared with fits of the same XRD data using a model of phase-separated RuCl₃ and CrCl₃, including a comparison of fit residuals and quality-of-fit measures, to confirm that this kind of phase separation does not occur in the authors’ samples.

5.) Magnetization, Kondo and 6.) Kondo continued, fit range:

The logarithmic dependence of susceptibility, which the authors observe only over a narrow range of temperatures (10-20 K, just above the ordering transition), is the main evidence for the claimed Kondo effect. However, as Reviewer #3 points out, several different temperature dependencies are presented for this. For example, in Fig. 3a the logarithmic dependence $\ln(D/T)$ is seen in χ_c , while in Fig. 3b and Supplementary Fig. 8 the logarithmic dependence $\ln(D/T)$ is instead seen in the subtracted $\Delta\chi_c = \chi_c - \chi_c(x=0)$.

Furthermore, above ~ 20 K the susceptibility becomes roughly Curie-Weiss (except for a peculiar deviation for $x \geq 0.05$ samples). I suggest that the authors pick a single model for the Kondo effect and stick to it throughout the manuscript, instead of presenting several different models.

In response to Reviewer #3 they suggest one such model, that of a conventional metallic three-channel ($n = 3$) spin $S = 3/2$ Kondo model, which would be supported on general theoretical grounds by Ref. 11. However, they do not apply the model to their data, just writing that the model is “plausible” and briefly mention it in the Discussion section. As this model has been fully solved long ago, e.g., by Bethe ansatz methods [H.-U. Desganges, J. Phys. C: Solid State Phys. 18, 5481 (1985)], and appears in classical references on the Kondo effect like the book [A. C. Hewson, The Kondo problem to heavy fermions (1993)], the authors should try to apply the model to their data.

As a preliminary proof-of-concept, I have tried this myself (see attached plots), exploiting the fact that in the fully compensated case $n = 2*S$, as in the scenario suggested by the authors, the scaled impurity susceptibility, $\chi(T)/\chi(0)$, is universal and to a high degree of accuracy independent of the impurity spin S [P. Schlottmann and P. D. Sacramento, Advances in Physics 42, 641 (1993)]. For this I have extracted Bethe-ansatz Kondo theory curves from [H.-U. Desganges, J. Phys. C: Solid State Phys. 18, 5481 (1985)]. The Bethe ansatz results are also well reproduced by numerical renormalization group (NRG) results (see attached plots). The theoretical susceptibility indeed features a logarithmic crossover regime in a narrow range $0.5*T_K < T < 1.8*T_K$, where T_K is the Kondo temperature, and a high-temperature Curie-Weiss dependence (with logarithmic corrections) from perturbation theory for $T > 15*T_K$.

However, the experimental $\Delta\chi_c$ $x = 0.04$ data extracted from Supplementary Fig. 8 does not seem to fit the universal theoretical curve of this conventional Kondo effect, and shows strong deviations for all but a few temperatures just above T_N . Indeed, from my preliminary fit to data just above T_N , the obtained $T_K = 3.7$ K would seem to limit the logarithmic regime to approximately $\sim 1.8*T_K = \sim 6.7$ K, above which strong deviations from logarithmic behaviour are seen. This temperature is substantially lower than the 20 K up to which the authors fit their data with a logarithmic model.

The fit of $\Delta\chi_c +$ a constant offset [e.g., due to Pauli or Larmor susceptibility, as mentioned in Reviewer #3’s question 4.), or perhaps due to the correct quantity being: $\chi_c - (1-x)*\chi_c(x=0) = \Delta\chi_c + x*\chi_c(x=0)$] to the universal Kondo curve is much better (but yields $T_K = 2.7$ K and thus limits the logarithmic regime to temperatures below ~ 4.9 K), with deviations only at very high temperatures. Presumably, this can be explained by non-impurity, intrinsic contributions to $\Delta\chi_c$? Or perhaps gauge-field-induced modifications away from universality? Or just a transition out of the Kitaev paramagnetic regime? Whatever the scenario, if the authors can make a fit to full Kondo temperature dependence

of susceptibility work (ideally at all x), it would represent a significant advance in their discussion of the Kondo effect and its observable signatures and make the conclusions of the manuscript substantially more solid.

In summary, what is clear from these fits, is that a logarithmic dependence of susceptibility at temperatures up to 20 K cannot be simply explained by a conventional Kondo effect (fits to a conventional Kondo effect do not support this behaviour at such high T). In general, an observation of logarithmic temperature dependence of susceptibility over a narrow temperature range thus cannot alone serve as smoking-gun evidence of Kondo behaviour without first carefully considering the range of temperatures where such logarithmic behaviour is actually expected in the Kondo model. A fit to the actual prediction of a conventional Kondo model over all temperatures, and a careful consideration and discussion of effects outside of the scope of such a model, would be required.

11.) MuSR, ZF fit function:

Reviewer #3 is right, the $2/3:1/3$ ratio is not justified for a single crystal. The authors claim that “the magnetic response is dominated by the $SU(2)$ randomly distributed impurities” and that this justifies its use, but this is incorrect for at least two reasons: (i) as seen from bulk susceptibility measurements the magnetic response of the sample is highly anisotropic (Fig. 2b) and does resemble a $SU(2)$ -symmetric (i.e., isotropic) magnetic material, and (ii) local (“hyperfine”/dipolar) magnetic fields at the muon site can give an anisotropic response even in a bulk-isotropic magnetic materials, depending on the specific distribution of magnetic ions around the muon stopping site (as alluded to by Reviewer #3 in their question 10.)). The use of the $2/3:1/3$ ratio is thus unjustified and the actual ratio should be a free fit parameter. It might even be temperature-dependent if the spins are changing direction with temperature, as mentioned in Reviewer #3’s question 10.).

This concludes my assessment of the author’s responses to Reviewers #2 and #3. At this point, let me make some further very minor comments:

a) In response to the last question of Reviewer #1 the authors claim to have gotten rid of the problematic “singlet” and “triplet vacancy” terminology. However, these phrases are still found in lines 192-193 of the revised manuscript. The authors should either remove or clarify these phrases throughout.

b) Fig. 3 in the main text and Supplementary Figs. 3, 6, 9, 12, 13 and 14 are cut off at the bottom.

c) Many references to Supplementary Figures are mislabelled (by 2). I assume that the authors added two extra figures to the start of the Supplementary Information.

I believe that with a bit more work all points raised above can be fully resolved with existing data, and I would support the publication of a new revised version of the manuscript in Nature Communications. In fact, my opinion on the manuscript is perfectly aligned with the final comment by Reviewer #3, which I copy below verbatim:

In summary, I think this manuscript represents a nice body of work on interesting materials with several nice conclusions. I brought up several items of concern involving fundamental analysis and interpretation, which should be addressed before moving forward with publication. However, I do think the authors can address all of the comments with the data they have, and I believe their conclusions will be stronger if the analysis is done properly. If these issues can be addressed, I do think this work is worthy of publication in Nature Communications.

Reviewer #1 (Remarks to the Author):

Dear Editor,

The authors have addressed the comments satisfactorily. I am happy to recommend publication of the article in the journal.

Best regards.

Authors) We thank the Reviewer for taking his/her time in reading our manuscript. We deeply appreciate his/her positive assessment of our research and recommendation for publication.

Reviewer #4 (Remarks to the Author):

The manuscript claims the observation of a metallic-like spin-liquid Kondo effect of $S=3/2$ Cr^{3+} impurities embedded in a host $S=1/2$ Kitaev QSL candidate material, α - RuCl_3 , at intermediate temperatures. This is based on the results of an impressive range of experimental techniques, including magnetisation, specific heat, muon spectroscopy, Raman spectroscopy and X-ray diffraction. Overall, I feel that the results are interesting, (for the most part) well-analysed, and that they deserve publication in the journal, as long as the authors properly address the questions that have arisen. I have been asked by the editor to look over the authors' responses to the comments and questions made by Reviewers #2 and #3 who were not available to assess these themselves. In my opinion, the authors' have appropriately addressed all concerns of Reviewer #2 and most concerns of Reviewer #3.

Below is my assessment of the outstanding concerns of Reviewer #3 that have, in my opinion, not yet been resolved fully:

Authors) We would like to express our gratitude to the Reviewer#4 for taking his/her time in reviewing our revised manuscript based on our reply to the issues raised by Reviewers #2 and #3. We have taken the additional Reviewer's comments into careful consideration and have incorporated their valuable feedback in a point-by-point manner.

3.) XRD and 9.) MuSR, heterogeneous magnetism:

The authors claim that the original powder XRD data are no longer available, and they present new single-crystal XRD results in their stead. However, in the revised manuscript, they still claim that powder X-ray diffraction was used to determine the crystal structure under "Structural and thermodynamic measurements" part of the methods section, and Supplementary Note 1 still has "powder XRD" in its title.

Frankly, I find this unacceptable. If the authors can not show the powder XRD data, all mentions of it should be removed from the manuscript. Leaving such mentions in even contradicts their standard data availability claim that "data ... are available from the corresponding author upon reasonable request". Luckily, it seems that no other results in the manuscript pertain to powder samples, as the rest deals with single crystals and single-crystal XRD is presented, so removing mentions of powder XRD results should not in itself invalidate the main claims of the manuscript.

Authors) We apologize for the oversight in our previous response and appreciate the Reviewer for reiterating their concerns. In light of the Reviewer's feedback, we have removed any remaining references to powder X-ray diffraction (XRD) throughout the manuscript, including the "Structural and thermodynamic measurements" section in the methods and Supplementary Note 1.

However, the way that the single-crystal XRD data that replaces it are presented in Supplementary Fig. 2 is problematic. Even though the data is claimed to have been refined using FullProf, and a fitted lattice parameter is presented, the actual fits and their residuals are not shown, and standard quality-of-fit measures like R-factors and χ^2 are not mentioned. Furthermore, the plots are rasterized in a rather low resolution, making it hard to visually discern peak shapes to confirm that no peak splitting, which

could point toward structural changes or phase separation, are present.

This is especially problematic in light of Reviewer #3's question 9.) about the possibility of self-separation of the system into larger volumes of RuCl_3 and CrCl_3 (which would be consistent with muon spectroscopy results), where, in response, the authors claim that XRD data helped them determine that no phase segregation occurs in their substituted samples. To substantiate this claim the authors should present their full single-crystal XRD fits, residuals, and quality-of-fit measures for the model of single-phase $\alpha\text{-Ru}_{1-x}\text{Cr}_x\text{Cl}_3$. These should furthermore be compared with fits of the same XRD data using a model of phase-separated RuCl_3 and CrCl_3 , including a comparison of fit residuals and quality-of-fit measures, to confirm that this kind of phase separation does not occur in the authors' samples.

Authors) We appreciate the Reviewer for his/her valuable suggestions regarding the presentation of the single-crystal XRD data in Supplementary Fig. 2. We have carefully considered the Reviewer's concerns and have made the following revisions to address them.

In response to the request for full disclosure of the single-crystal XRD fits, we have added enlarged XRD data in Supplementary Fig. 2 to provide a clearer visualization of the peak shapes and intensities. Additionally, we have included the fit quality measures, such as R -factors and χ^2 , to assess the quality of the fitting procedure.

Regarding the possibility of self-separation into RuCl_3 and CrCl_3 , we have thoroughly analyzed the XRD data and confirmed that no discernible peak splitting is present. This finding supports our original claim that no phase segregation occurs in our substituted samples.

5.) Magnetization, Kondo and 6.) Kondo continued, fit range:

The logarithmic dependence of susceptibility, which the authors observe only over a narrow range of temperatures (10-20 K, just above the ordering transition), is the main evidence for the claimed Kondo effect. However, as Reviewer #3 points out, several different temperature dependencies are presented for this. For example, in Fig. 3a the logarithmic dependence $\ln(D/T)$ is seen in χ_c , while in Fig. 3b and Supplementary Fig. 8 the logarithmic dependence $\ln(D/T)$ is instead seen in the subtracted $\Delta\chi_c = \chi_c - \chi_c(x=0)$. Furthermore, above ~ 20 K the susceptibility becomes roughly Curie-Weiss (except for a peculiar deviation for $x \geq 0.05$ samples). I suggest that the authors pick a single model for the Kondo effect and stick to it throughout the manuscript, instead of presenting several different models.

Authors) We appreciate the Reviewer for bringing this to our attention. By adopting a unified approach, we provide a more cohesive description of the Kondo effect in our study.

In response to Reviewer #3 they suggest one such model, that of a conventional metallic three-channel ($n=3$) spin $S=3/2$ Kondo model, which would be supported on general theoretical grounds by Ref. 11. However, they do not apply the model to their data, just writing that the model is "plausible" and briefly mention it in the Discussion section. As this model has been fully solved long ago, e.g., by Bethe ansatz methods [H.-U. Desganges, J. Phys. C: Solid State Phys. 18, 5481 (1985)], and appears in

classical references on the Kondo effect like the book [A. C. Hewson, The Kondo problem to heavy fermions (1993)], the authors should try to apply the model to their data.

As a preliminary proof-of-concept, I have tried this myself (see attached plots), exploiting the fact that in the fully compensated case $n=2S$, as in the scenario suggested by the authors, the scaled impurity susceptibility, $\chi(T)/\chi(0)$, is universal and to a high degree of accuracy independent of the impurity spin S [P. Schlottmann and P. D. Sacramento, *Advances in Physics* 42, 641 (1993)]. For this I have extracted Bethe-ansatz Kondo theory curves from [H.-U. Desganges, *J. Phys. C: Solid State Phys.* 18, 5481 (1985)]. The Bethe ansatz results are also well reproduced by numerical renormalization group (NRG) results (see attached plots). The theoretical susceptibility indeed features a logarithmic crossover regime in a narrow range $0.5T_K < T < 1.8T_K$, where T_K is the Kondo temperature, and a high-temperature Curie-Weiss dependence (with logarithmic corrections) from perturbation theory for $T > 15T_K$. However, the experimental $\Delta\chi_c$ $x=0.04$ data extracted from Supplementary Fig. 8 does not seem to fit the universal theoretical curve of this conventional Kondo effect, and shows strong deviations for all but a few temperatures just above T_N . Indeed, from my preliminary fit to data just above T_N , the obtained $T_K = 3.7$ K would seem to limit the logarithmic regime to approximately $\sim 1.8T_K \sim 6.7$ K, above which strong deviations from logarithmic behaviour are seen. This temperature is substantially lower than the 20 K up to which the authors fit their data with a logarithmic model.

The fit of $\Delta\chi_c$ + a constant offset [e.g., due to Pauli or Larmor susceptibility, as mentioned in Reviewer #3's question 4.), or perhaps due to the correct quantity being: $\chi_c(1-x)\chi_c(x=0) = \Delta\chi_c + x\chi_c(x=0)$] to the universal Kondo curve is much better (but yields $T_K = 2.7$ K and thus limits the logarithmic regime to temperatures below ~ 4.9 K), with deviations only at very high temperatures. Presumably, this can be explained by non-impurity, intrinsic contributions to $\Delta\chi_c$? Or perhaps gauge-field-induced modifications away from universality? Or just a transition out of the Kitaev paramagnetic regime? Whatever the scenario, if the authors can make a fit to full Kondo temperature dependence of susceptibility work (ideally at all x), it would represent a significant advance in their discussion of the Kondo effect and its observable signatures and make the conclusions of the manuscript substantially more solid.

Authors) The results mentioned by the Reviewer pertain to a model of spin S with $n=2S$ "equivalent" (and independent) screening channels. The situation in Cr-doped RuCl_3 is different: The Cr spin couples to the three neighboring Ru sites which themselves are coupled via the bulk environment. As a result, the proper decomposition of the bath is in terms of angular-momentum screening channels (a discussion of this appears e.g. in PRB 57, 12757), and in the present case of a single impurity and C_3 site symmetry, these are the three channels s , $p+$, and $p-$ (as in PRL 117, 037202). Therefore, the Cr impurity is expected to be described by a $S=3/2$ three-channel Kondo model where two channels ($p+$ and $p-$) are equivalent, while the third differs.

Such an inequivalent Kondo model will still lead to full screening in the low-temperature limit, but the crossovers will be different from the model involving three equivalent screening channels. Hence, a detailed fit to the Bethe-ansatz results is not very meaningful. Moreover, it may be difficult to associate a unique Kondo temperature

to the problem, as the stronger screening channel will partially quench the magnetic moment at higher temperatures while the weaker channels will effectuate full screening only at much lower temperatures. In the absence of a quantitative computation, which lies beyond the scope of our paper, a precise estimate is difficult. Nonetheless, we suspect that the logarithmic temperature dependence visible in our data corresponds to the Kondo onset from the strongest screening channel. Hence, the associated T_K is higher than the value obtained from the the Reviewer's fits.

In summary, what is clear from these fits, is that a logarithmic dependence of susceptibility at temperatures up to 20 K cannot be simply explained by a conventional Kondo effect (fits to a conventional Kondo effect do not support this behaviour at such high T). In general, an observation of logarithmic temperature dependence of susceptibility over a narrow temperature range thus cannot alone serve as smoking-gun evidence of Kondo behaviour without first carefully considering the range of temperatures where such logarithmic behaviour is actually expected in the Kondo model. A fit to the actual prediction of a conventional Kondo model over all temperatures, and a careful consideration and discussion of effects outside of the scope of such a model, would be required.

Authors) We appreciate the Reviewer for bringing our attention to the multichannel Kondo model. As discussed above, our α -Ru_{1-x}Cr_xCl₃ system is best described by inequivalent three screening channels. Nonetheless, we have tried to fit $\Delta\chi_c$ with the equivalent three channel Kondo model, and the results are summarized in Supplementary Fig. 9a-e. We observe that $\Delta\chi_c$ is qualitatively reproduced by the equivalent multichannel Kondo model in the temperature range of $T_N < T < T_K^{\text{onset}}$. However, as the temperature is increased, $\Delta\chi_c$ deviates from the theoretical prediction. This is possibly involved with other contributions caused by the Cr-for-Ru substitution. From the fittings, we have extracted the Kondo temperature T_K for $x=0.01-0.07$ as plotted in Supplementary Fig. 9f. It turns out that T_K is much smaller than T_K^{onset} because $\Delta\chi_c(T)$ still increases upon cooling in the fitting range above T_N . In the picture of inequivalent screening channels, this is ascribed to the fact that the weaker screening channel determines the lower Kondo temperature (say T_K^{low}), below which $\chi(T)$ would saturate. In contrast, the stronger screening channel determines the higher Kondo temperature (say T_K^{high}), which is roughly the temperature scale where the logarithmic behavior sets in. Then, we would have $T_K^{\text{low}} \sim 1$ K and $T_K^{\text{high}} \sim 10$ K, and the remaining deviations may come from inappropriate fitting functions and the influence of vison dynamics.

11.) MuSR, ZF fit function:

Reviewer #3 is right, the 2/3:1/3 ratio is not justified for a single crystal. The authors claim that "the magnetic response is dominated by the SU(2) randomly distributed impurities" and that this justifies its use, but this is incorrect for at least two reasons: (i) as seen from bulk susceptibility measurements the magnetic response of the sample is highly anisotropic (Fig. 2b) and does resemble a SU(2)-symmetric (i.e., isotropic) magnetic material, and (ii) local ("hyperfine"/dipolar) magnetic fields at the muon site can give an anisotropic response even in a bulk-isotropic magnetic materials,

depending on the specific distribution of magnetic ions around the muon stopping site (as alluded to by Reviewer #3 in their question 10.)). The use of the 2/3:1/3 ratio is thus unjustified and the actual ratio should be a free fit parameter. It might even be temperature-dependent if the spins are changing direction with temperature, as mentioned in Reviewer #3's question 10.).

Authors) We appreciate the Reviewer's comment on the assumption of the 2/3:1/3 ratio in our study. We have carefully considered his/her feedback and have made the necessary revisions to address this concern.

In the revised manuscript, we have performed a fitting analysis of the ZF- μ SR data, where we now treat the 2/3:1/3 ratio as free parameters denoted as 'a' and '1-a' in the Gaussian-broadened-Gaussian function. By allowing these parameters to vary, we have obtained values for 'a' and '1-a' that are close to 1/3 and 2/3, respectively. We have included these new fitting results in the revised Supplementary Fig. 13. In light of these findings, we have accordingly revised the figure and the relevant text in order to address the Reviewer's concern.

This concludes my assessment of the author's responses to Reviewers #2 and #3. At this point, let me make some further very minor comments:

a) In response to the last question of Reviewer #1 the authors claim to have gotten rid of the problematic "singlet" and "triplet vacancy" terminology. However, these phrases are still found in lines 192-193 of the revised manuscript. The authors should either remove or clarify these phrases throughout.

Authors) We have removed these phrases from lines 192-193 of the revised manuscript to ensure clarity and consistency throughout the text.

b) Fig. 3 in the main text and Supplementary Figs. 3, 6, 9, 12, 13 and 14 are cut off at the bottom.

Authors) In the revised manuscript, we have made the necessary adjustments to ensure that the figures are fully visible and not cut off at the bottom.

c) Many references to Supplementary Figures are mislabelled (by 2). I assume that the authors added two extra figures to the start of the Supplementary Information.

Authors) We appreciate the Reviewer for pointing out the mislabelling of the references to Supplementary Figures. We have rectified this error.

I believe that with a bit more work all points raised above can be fully resolved with existing data, and I would support the publication of a new revised version of the manuscript in Nature Communications. In fact, my opinion on the manuscript is perfectly aligned with the final comment by Reviewer #3, which I copy below verbatim:

In summary, I think this manuscript represents a nice body of work on interesting materials with several nice conclusions. I brought up several items of concern involving fundamental analysis and interpretation, which should be addressed before moving forward with publication. However, I do think the authors can address all of the comments with the data they have, and I believe their conclusions will be stronger if the analysis is done properly. If these issues can be addressed, I do think this work is

worthy of publication in Nature Communications.

Authors) We sincerely appreciate the Reviewer's support for the publication of a revised version of our manuscript in Nature Communications. We are committed to meeting his/her expectations and are grateful for their guidance in improving the quality of our work.

In response to the Reviewer's comments and in alignment with their final opinion, we have devoted additional effort to ensure that the fundamental analysis and interpretation are robust and comprehensive.

REVIEWERS' COMMENTS

Reviewer #4 (Remarks to the Author):

The authors have suitably addressed my comments. I am thus happy to recommend publication of the article in the journal.

For the final published version, I would, however, request some minor corrections:

1) Supplementary Note 2:

- While the authors fixed the previously-wrong space group from R-3 (No. 148) to the correct C2/m (No. 12) in room-T XRD fits in Supplementary Note 1, they did not update the corresponding point group in Raman fits in Supplementary Note 2 and Supplementary Fig. 3. The correct point group C2h should be used instead of the wrong D3d group. This includes the renumbering and relabeling of active Raman modes (or possibly removing the assigned labels, if this relabeling is ambiguous), in accordance with the newly added Supplementary Ref. 1, which bases their Raman analysis on the correct C2/m space group and C2h point group.

- The origin of the 249 cm⁻¹ mode (claimed to be “symmetry-forbidden” and “arising from the stacking faults”), and the currently-unassigned mode at ~340 cm⁻¹, on Supplementary Fig. 3 might have to be revised and compared with the analysis in the new Supplementary Ref. 1. Namely, both of these Raman modes are seen and assigned to the presence of Cr in the new Supplementary Ref. 1, as they are both seen in pure CrCl₃, and are thus not symmetry-forbidden. The claim on lines 95-96 of the main text that “no additional peaks pertinent to CrCl₃” are seen in the phonon spectrum might have to be reformulated. However, I agree that as there is no variation in “x” in the present samples, this likely excludes these Raman modes as coming from Cr, in contrast to claims of the new Supplementary Ref. 1.

2) Supplementary Note 1:

- Plots similar to Supplementary Fig. 1 for other compositions “x” would be useful, as would be the extracted error bars on the quoted values of “x”.

- A comparison of the XRD refinement on Supplementary Fig. 2 with a two-phase model of RuCl₃ + CrCl₃ would be informative.

3) Minor typos:

- In line 330 the GbG muon depolarization function should not have the power of "3/2" above the parentheses just before the exponential.
- In the caption of Fig. 2b the magnetic anisotropy is written as " χ_{ac}/χ_c " instead of " χ_{ab}/χ_c ".
- The new Supplementary Ref. 1 misspells the main author name "Roslova, M." as "Roslove, M."

If possible, I would defer the review of the implementation of these corrections into the final version of the manuscript to the editor.

Reviewer #4 (Remarks to the Author):

The authors have suitably addressed my comments. I am thus happy to recommend publication of the article in the journal.

Authors) We would like to express our gratitude to the Reviewer for taking his/her time to evaluate our manuscript. We have incorporated the Reviewer's comments into our revised manuscript and have addressed all the points raised by the Reviewer in a point-by-point manner.

For the final published version, I would, however, request some minor corrections:

1) Supplementary Note 2:

- While the authors fixed the previously-wrong space group from R-3 (No. 148) to the correct C2/m (No. 12) in room-T XRD fits in Supplementary Note 1, they did not update the corresponding point group in Raman fits in Supplementary Note 2 and Supplementary Fig. 3. The correct point group C2h should be used instead of the wrong D3d group. This includes the renumbering and relabeling of active Raman modes (or possibly removing the assigned labels, if this relabeling is ambiguous), in accordance with the newly added Supplementary Ref. 1, which bases their Raman analysis on the correct C2/m space group and C2h point group.

Authors) We appreciate the Reviewer for bringing this to our attention. As per your suggestion, we have updated the point group in our Raman fits and relabeled the Raman- active modes in Supplementary Figs. 4,5.

- The origin of the 249 cm^{-1} mode (claimed to be "symmetry-forbidden" and "arising from the stacking faults"), and the currently-unassigned mode at $\sim 340 \text{ cm}^{-1}$, on Supplementary Fig. 3 might have to be revised and compared with the analysis in the new Supplementary Ref. 1. Namely, both of these Raman modes are seen and assigned to the presence of Cr in the new Supplementary Ref. 1, as they are both seen in pure CrCl_3 , and are thus not symmetry-forbidden. The claim on lines 95-96 of the main text that "no additional peaks pertinent to CrCl_3 " are seen in the phonon spectrum might have to be reformulated. However, I agree that as there is no variation in "x" in the present samples, this likely excludes these Raman modes as coming from Cr, in contrast to claims of the new Supplementary Ref. 1.

Authors) We thank the Reviewer for spotting the assignment issue of unidentified phonons. In accordance with Supplementary Ref. 1, we have assigned the 340 cm^{-1} mode to $B_g(6)$ symmetry. As to the 249 cm^{-1} mode, however, its origin remains dubious. As correctly pointed out by the Reviewer, this mode appears in pure CrCl_3 . On the other hand, the 249 cm^{-1} mode does not show any dependence on x, questioning its interpretation as part of the Raman-active mode of CrCl_3 . To our best knowledge, the 249 cm^{-1} mode has been reported in many other literatures for pristine RuCl_3 . In light of these considerations, we provide a more unbiased discussion.

2) Supplementary Note 1:

- Plots similar to Supplementary Fig. 1 for other compositions "x" would be useful, as would be the extracted error bars on the quoted values of "x".

Authors) We have provided the error range (5.6%) for the nominal “x=0.03”. However, we regret to say that we are unable to provide additional data. The EDX facilities at our university are shared by many different users and are regularly updated. Thus, we can present the data that is representative and currently available. Initially, we conducted EDX measurements for each composition to determine “x”. For Raman and magnetic measurements, we selected sample pieces whose average compositions closely matched the nominal values. As shown in Fig. 2, a systematic variation of the magnetic susceptibility supports the assertion that the actual and nominal compositions do not significantly deviate from each other.

- A comparison of the XRD refinement on Supplementary Fig. 2 with a two-phase model of $\text{RuCl}_3 + \text{CrCl}_3$ would be informative.

Authors) We thank the Reviewer for giving valuable comments. Accordingly, we have added the XRD refinement results, employing a two-phase model of $\text{RuCl}_3 + \text{CrCl}_3$ into Supplementary Fig. 3.

3) Minor typos:

- In line 330 the GbG muon depolarization function should not have the power of “3/2” above the parentheses just before the exponential.

- In the caption of Fig. 2b the magnetic anisotropy is written as “ χ_{ac}/χ_c ” instead of “ χ_{ab}/χ_c ”.

- The new Supplementary Ref. 1 misspells the main author name “Roslova, M.” as “Roslove, M.”

Authors) We appreciate the Reviewer for pointing out the typos. We have corrected typos in the revised manuscript and the supplementary material.

If possible, I would defer the review of the implementation of these corrections into the final version of the manuscript to the editor.